



# 1 Traffic-related air pollution near roadways: discerning local impacts
# 2 from background

Nathan Hilker[1], Jonathan M. Wang[1], Cheol-Heon Jeong[1], Robert M. Healy[2], Uwayemi Sofowote[2], Jerzy
Debosz[2], Yushan Su[2], Michael Noble[2], Anthony Munoz[2], Geoff Doerksen[3], Luc White[4], Céline Audette[4],
Dennis Herod[4], Jeffrey R. Brook[1,5], Greg J. Evans[1]
[1]Southern Ontario Centre for Atmospheric Aerosol Research, Department of Chemical Engineering and Applied Chemistry,
University of Toronto, Toronto, ON, M5S 3E5, Canada
[2]Environmental Monitoring and Reporting Branch, Ontario Ministry of the Environment Conservation and Parks, Etobicoke,
ON, M3P 3V6, Canada
[3]Air Quality Policy and Management Division, Metro Vancouver, Burnaby, BC, V5H 0C6, Canada
[4]Air Quality Research Division, Environment and Climate Change Canada, Ottawa, ON, K1A 0H3, Canada
[5]Air Quality Research Division, Environment and Climate Change Canada, Toronto, ON, M3H 5T4, Canada
*Correspondence to*: Greg J. Evans (greg.evans@utoronto.ca)



**Abstract.** Adverse health outcomes related to exposure to air pollution have gained much attention in recent years, with a particular emphasis on traffic-related pollutants near roadways, where concentrations tend to be most severe. As such, many projects around the world are being initiated to routinely monitor pollution near major roads. Understanding the extent to which local on-road traffic directly affects these measurements, however, is a challenging problem, and a more thorough comprehension of it is necessary to properly assess its impact on near-road air quality. In this study, a set of commonly measured air pollutants (black carbon; carbon dioxide; carbon monoxide; fine particulate matter, $PM_{2.5}$; nitrogen oxides; ozone; and ultrafine particle concentrations) were monitored continuously between June 01st, 2015 and March 31st, 2017 at six stations in Canada: two near-road and two urban background stations in Toronto, Ontario, and one near-road and one urban background station in Vancouver, British Columbia. Three methods of differentiating between local and background concentrations at near-road locations were tested: 1) differences in average pollutant concentrations between near-road and urban background station pairs, 2) differences in downwind and upwind pollutant averages, and 3) interpolation of rolling minima to infer background concentrations. The latter two methods use near-road data only, and were compared with method 1, where an explicit difference was measured, to assess accuracy and robustness. It was found that method 2 produced average local concentrations that were biased high by a factor of between 1.4 and 1.7 when compared with method 1 and was not universally feasible, whereas method 3 produced concentrations that were in good agreement for all pollutants except ozone and $PM_{2.5}$, which are generally secondary and regional in nature. The results of this comparison are intended to aid researchers in the analysis of data procured in future near-road monitoring studies. Lastly, upon determining these local pollutant concentrations as a function of time, their variability with respect to wind speed (WS) and wind direction (WD) was assessed. With the exception of ozone and $PM_{2.5}$, local pollutant concentrations were enhanced by a factor of 2 relative to their mean in the case of stagnant winds and were shown to be proportional to $WS^{-0.6}$. Downwind conditions enhanced local concentrations by a factor of ~2 relative to their mean, while upwind conditions suppressed them by a factor of ~4.



# 1 Introduction

Exposure to elevated air pollutant concentrations is an ongoing concern as it has been identified as a risk factor for a variety of adverse health outcomes, including: cardiovascular disease (Brook et al., 2004), increased arterial calcification (Kaufman et al., 2016) and myocardial infarction incidence rates (Wolf et al., 2015); vascular dementia, Alzheimer's disease (Oudin et al., 2016), and impaired brain development in children (Clifford et al., 2016); chronic obstructive pulmonary disease (Andersen et al., 2011), asthma in both adults and children (Brauer et al., 2007; Kunzli et al., 2009), and lung cancer (Hamra et al., 2015). Lelieveld et. al (2015) estimated that exposure to air pollution resulted in 3.3 million premature deaths globally in 2010, with land traffic sources being a primary contributor in North America and some European countries. Traffic-related air pollutants (TRAPs) are of concern, because on-road traffic is often a major source of air pollution in urban environments (Belis et al., 2013; Molina and Molina, 2004; Pant and Harrison, 2013) where population densities are greatest—in Canada, it is estimated that one third of the population live within 250 m of a major roadway (Evans et al., 2011)—and it is within these near-road regions TRAP concentrations are generally highest (Baldwin et al., 2015; Jeong et al., 2015; Kimbrough et al., 2018; Saha et al., 2018).

As such, there is a growing interest in measuring air pollutant concentrations near roadways in order to better understand TRAP exposure levels in these environments. However, in order to understand the underlying sources and reasons for elevated concentrations, further processing of raw measurement data is necessary. In general, near-road TRAP concentrations are influenced by both regional and local emissions, and being able to distinguish the contributions of these sources allows their relative impacts to be more properly assessed. Of particular importance to near-road measurements is understanding the role of on-road traffic. For TRAPs whose source(s) cannot be readily identified from their measurement at a singular location, concurrent samples at various locations and/or algorithmic methods can be used to enable apportionment.

Often, determining TRAP background concentrations is accomplished through monitoring at remote, representative locations that are minimally impacted by nearby sources; properly siting background stations in urban environments is in itself a challenge, and not always feasible. This practice, while useful in providing confidence in information regarding background air quality, is expensive because it requires additional monitoring stations and personnel to maintain them. The value of these background stations is lessened if similar knowledge is extractable from near-road locations alone. Various time-series analysis algorithms have been proposed for this purpose, many of which make use of the inverse relation between source proximity and signal frequency. For example, work by Klems et al., (2010) and Sabaliauskas et al., (2014) made use of the discrete wavelet transform, an algorithm used widely in signal compression and denoising, to ultrafine particle time-series data to determine the time-dependent contribution of local sources to roadside concentrations. Additionally, the technique of interpolating minima across time windows of varying length has been applied successfully to data from both mobile laboratories (Brantley et al., 2014; Shairsingh et al., 2018) and stationary measurements (Wang et al., 2018) for the purposes of estimating urban background pollutant concentrations.



Given the diversity of techniques available for differentiating local and background pollutant concentrations, as well as the
large variety of instrumentation available, it is not clear which approaches are most generalizable or applicable, or whether it
is necessary to invest in concurrent measurements at many versus few locations. In addition, the exact definition of what is
background air quality is somewhat unclear, and in the context of this study, given the spatial separation between sites (on the
order of 10 km or less), it is assumed to be a measure of background air quality in the urban airshed. Ma and Birmili (2015),
in a study of ultrafine particle nucleation, for example, defined measurement locations in their study which were 4.5 km and
40 km from an urban roadside station as urban background and regional background, respectively. The former was presumed
to be a measure of regional air quality superimposed with diffuse urban emissions, and it is this definition that best characterizes
the background air quality measured in this study. To evaluate whether information regarding this urban background was
attainable from near-road measurements alone, two strategies for quantifying the contribution of local on-road traffic to near-
road air quality were compared, and their reliability and accuracy were assessed through comparison with tandem
measurements in both environments.
In this study, data were collected continuously at three near-road and three urban background monitoring locations for close
to two years (namely, between June 01st, 2015, and March 31st, 2017). Various gas and particle-phase pollutants along with
meteorological parameters were measured using an array of instrumentation. Concentrations in excess of the urban background
were calculated from the near-road data using three techniques, one of which calculated an explicit difference between sites,
whereas the other two made use of only near-road data. Comparison of these methodologies addresses whether information
regarding background air quality is readily inferable from measurements made in the near-road environment.

## 2 Methods

### 2.1 Measurement locations

Data were collected from six separate monitoring locations: four of which were in Toronto, Ontario (two situated near
roadways and two in urban background environments), with the remaining two located in Vancouver, British Columbia (one
situated near a roadway and another in the urban background). The location of each station, along with information regarding
the major roadway next to which they were located (for the near-road sites), is summarized in Table 1. The two near-road
stations in Toronto, NR-TOR-1 (43.7111, -79.5433) and NR-TOR-2 (43.6590, -79.3954), and their respective instrumentation
setups have been utilized and reported by others and are described therein (Sabaliauskas et al., 2012; Sofowote et al., 2018;
Wang et al., 2015). The NR-TOR-1 site was positioned 10 m from Highway 401, the busiest highway in North America in
terms of Annual Average Daily Traffic (AADT) with over 400,000 vehicles per day distributed across eight eastbound and
eight westbound lanes. The Southern Ontario Centre for Atmospheric Aerosol Research (SOCAAR) served as the second near-
road site (NR-TOR-2), and was located 15 m from College Street in downtown Toronto which experienced traffic volumes of
17,200 vehicles per day on average. The northernmost station in Toronto, BG-TOR-1, was located at Environment and Climate
Change Canada (43.7806, -79.4675), 180 m from the nearest roadway, and the measurements from this station served as an





urban background/baseline for NR-TOR-1, which was located 9.8 km to the southwest. The second background station, BG-
TOR-2, was positioned on the southernmost point of the Toronto Islands on Lake Ontario (43.6122, -79.3887), and was 5.2
km south of NR-TOR-2. Since vehicular traffic on the Toronto Islands was limited to a small number of service vehicles, the
BG-TOR-2 station was well removed from tailpipe emissions.
The near-road station in Vancouver, NR-VAN, was situated 6 m from Clark Drive (49.2603, -123.0778), a major roadway that
experienced on average 33,100 vehicles per day across four southbound and three northbound lanes. Additionally, located 65
m south of the station was a major intersection, Clark Drive and 12$^{th}$ avenue, at which there were two gas stations located on
the northwest and northeast sides. The effect this intersection had on traffic patterns (stop-and-go, especially) directly next to
the station, and its effect on measured TRAP concentrations are explored in this study. Lastly, the urban background station
in Vancouver, BG-VAN, was located 2.2 km east of NR-VAN at Sunny Hill Children's Hospital (49.2529, -123.0492). This
area was relatively removed from traffic emissions because it was located within a neighbourhood zoned predominately for
single unit family dwellings.
**2.2 Instrumentation**
A common suite of instrumentation was employed at all stations. Gas-phase pollutants measured include: carbon dioxide ($CO_2$;
840A, LI-COR Biosciences), carbon monoxide (CO; 48i, Thermo Scientific; attenuation of infrared radiation at a wavelength
of 4.6 µm), ozone ($O_3$; 49i, Thermo Scientific; attenuation of ultraviolet radiation at a wavelength of 254 nm), and nitrogen
oxides ($NO_x$; 42i, Thermo Scientific; infrared chemiluminescence). Particle-phase pollutant properties measured include: mass
concentration of particles less than 2.5 microns in diameter ($PM_{2.5}$; SHARP 5030, Thermo Scientific; beta attenuation and light
scattering); particle number concentration (UFP; 651, Teledyne API; condensation particle counter); and black carbon (BC;
AE33, Magee Scientific; attenuation of 880 nm wavelength light) mass concentration. Additionally, a meteorological sensor
(WXT520, Vaisala; ultrasonic anemometer) recorded wind direction, wind speed, ambient temperature, pressure, and relative
humidity at each station. Traffic intensities, velocities, and approximate vehicle lengths were measured continuously
(SmartSensor HD, Wavetronix; dual beam radar) at the three near-road stations.
Gas-phase instruments were calibrated on-site every two months using cylinders of compressed gasses at certified
concentrations (Linde). One cylinder contained $SO_2$, CO, and $CO_2$, while the other contained NO; both contained $N_2$ as an
inert makeup gas. Dilution and mixing of the gasses was accomplished using a dynamic gas calibrator (146i, Thermo Scientific)
to produce zero checks and span concentrations that were similar to ambient ranges. SHARP 5030 instruments were zero
checked using a HEPA filter, had their temperature and relative humidity sensors calibrated, and were span checked using
mass standards supplied by Thermo Fisher Scientific twice annually. In addition to recommended monthly maintenance
procedures for the API 651, each instrument underwent routine annual calibration by the manufacturer. Flow rates at each
station were verified on a monthly basis, and a variable flow rate pump was attached to a stainless steel particle manifold, from
which all particle-phase instruments sampled, to ensure a constant flow rate of 16.7 LPM to satisfy the 2.5 µm cut-off
conditions of the inlet cyclone.





## 3 Data analysis

Data acquisition was performed using Envidas Ultimate software (DR DAS Ltd.). Quality assurance of the data was performed by the primary operators of each station. This included, among other things: discounting data in which instrument diagnostic parameters were outside of acceptable ranges, omitting calibration times, and flagging suspect periods. Data from this study was acquired at a minutely resolution, and further averaged to hourly resolution. Only hours containing at least 45 minutes ($\geq$ 75%) of valid data are reported. Processing and analysis was accomplished through a combination of SQL (Microsoft), SAS 9.4 (SAS Institute Inc.), and IGOR Pro 6.37 (Wavemetrics Inc.) software. Using the hourly concentration in the finalized dataset, three methods of separating local and background concentrations from the near-road measurements were tested. One of these methods made use of the urban background measurements to explicitly infer background concentrations, whereas the other two, downwind/upwind comparison and time-series analysis, estimated background concentrations from the near-road measurements alone.

### 3.1 Average site differences

The first methodology for determining local pollutant concentrations explored in this paper, henceforth referred to as method 1, is through the difference between concentrations measured at a near-road location, $C_{NR}$, and at the nearest urban background location, $C_{BG}$, for some concurrent observation, i. Concentrations associated with local influences determined using method 1, $C_{L,1}$, rely on the assumption:

$$C_{NR}[i] = C_{L,1}[i] + C_{BG}[i]. \tag{1}$$

Average $C_{L,1}$ values for each near-road location were then determined using Eq. (2):

$$\bar{C}_{L,1} = \frac{1}{N} \sum_{i=1}^{N} C_{NR}[i] - C_{BG}[i], \tag{2}$$

again, $C_{NR}[i]$ and $C_{BG}[i]$ are near-road and urban background measurements, respectively, made over a concurrent time interval, i, As N, the number of observations used in calculating the temporal average increases, the calculated average difference is expected to converge to the true average difference between sites—a result encompassing more variability in meteorological and traffic conditions.

### 3.2 Downwind-upwind analysis

Through association with meteorology at a near-road measurement location, it is possible to assess traffic's influence on TRAP concentrations from the differences between downwind and upwind conditions. For example, Galvis et al. (2013) utilized average downwind and upwind concentrations of $CO_2$, BC, and $PM_{2.5}$ from a railyard to calculate local pollutant concentrations for use in fuel-based emission factor calculations. A similar approach is used here to isolate concentrations emitted from a roadway, henceforth referred to as method 2. Defining ranges of wind directions as corresponding to downwind and upwind





of the major street next to which a station is located, average local concentrations from method 2, $C_{L,2}$, can be estimated using
Eq. (3):
$$\bar{C}_{L,2} = \frac{1}{N}\sum_{i=1}^{N} C_{DW}[i] - \frac{1}{M}\sum_{i=1}^{M} C_{UW}[i] \,, \tag{3}$$
where $C_{DW}$ and $C_{UW}$ are near-road TRAP concentrations measured when winds originate from downwind and upwind of the
major roadway, respectively. Note that the number of points used to compute the averages of these conditions, N and M, need
not be equivalent, and the times that comprise these two averages are mutually exclusive by definition. Furthermore, similar
to method 1, as the averaging time for both conditions is increased, confidence in $C_{L,2}$ will improve. It is also important to note
that because these two meteorological scenarios encompass different time frames, it is possible for certain times of day, etc.
to be overrepresented in either average.
In all analysis in which meteorological data are utilized, stagnant periods (wind speed (WS) < 1.0 m s$^{-1}$) are omitted.
Furthermore, because wind direction may vary on the order of minutes, only hourly averages, generated from minutely data,
in which winds originated from a given sector over 75% of the time were deemed to be upwind or downwind. Local
concentrations cannot be estimated as a function of time using this method, as downwind and upwind concentrations cannot
be measured simultaneously with a single near-road station. Also, stagnant time periods, as well as time periods that are not
within the downwind/upwind ranges are omitted, thereby increasing the amount of time needed to attain a representative
average. Lastly, an inherent assumption to this method is that upwind concentrations on either side of the roadway are similar.
Depending on the site, however, this assumption may not be accurate.

### 3.2.1 Wind sector definitions at NR-TOR-1

Defining downwind and upwind sectors at NR-TOR-1 was straightforward, owing to the flat terrain of the area and the lack
of nearby TRAP sources excluding those from Highway 401. Hence, 90° quadrants perpendicular to the highway axis were
chosen. These definitions were further supported by average ambient $CO_2$ concentrations—an indicator of combustion
associated with traffic emissions—measured as a function of wind direction, shown in Fig. 1. Thus, downwind conditions at
NR-TOR-1 were defined as WD ≥ 295° or WD ≤ 25° and upwind as 115° ≤ WD ≤ 205°, where WD denotes wind direction
as measured locally at the station atop a 10 m mast.

### 3.2.2 Wind sector definitions at NR-TOR-2

Unlike the NR-TOR-1 site, wind dynamics at NR-TOR-2 were complicated by urban topography; namely, the roadside inlet
was within an urban canyon (aspect ratio of ~0.5: building heights of ~20 m on either side and a street width of ~40 m) resulting
in more stagnant conditions roadside and introducing micrometeorological effects such as in-canyon vortices (Oke, 1988). The
effect of urban canyon geometry on micrometeorology is an effect that has been known for some time, and in general, for city-
scale wind patterns perpendicular to the street axis, ground-level winds tend to be opposite with respect to the street axis to
those above the urban canopy (Vardoulakis et al., 2003).





Given the urban canyon's effect on ground-level wind direction, downwind/upwind quadrants at NR-TOR-2 were determined
based on wind direction measurements made above the urban canopy, and are defined as: WD ≥ 300° or WD ≤ 30° and 120°
≤ WD ≤ 210° for downwind and upwind conditions, respectively. Figure 2 shows a satellite image of the site with these
respective quadrant definitions, along with average $CO_2$ concentrations as a function of wind direction, similar to Fig. 1. From
the range of $CO_2$ concentrations seen here, it is clear that obtaining a precise definition of what exactly is downwind or upwind
of College Street is non-trivial. Impact from the intersection southwest (winds from ~230°) of the receptor is somewhat
apparent in Fig. 2.
**3.2.3 Wind sector definitions at NR-VAN**
While the presence of 2-3 story buildings within the immediate vicinity of the NR-VAN station may have complicated
meteorological measurements to some extent, the role of wind direction on the impact of local traffic emissions was much
more evident at this site than it was at NR-TOR-2. Other streets in the vicinity of Clark Drive affected the driving patterns near
the station—a major intersection (Clark Drive and 12[th] Avenue) approximately 65 m south of the station had an impact on
average measured $CO_2$ concentrations (Fig. 3) originating from the SSE direction. Because of this, the downwind and upwind
sector definitions for this site were not taken to be orthogonal: instead, downwind was defined as 135° ≤ WD ≤ 195° and
upwind as 235° ≤ WD ≤ 315°; these definitions were chosen in accordance with surrounding land usage. While the upwind
definition does include 12[th] avenue, a major roadway within 120 m of the station, it is suspected that lower TRAP
concentrations from this sector are due to: lower traffic volumes on 12[th] compared with Clark Drive, truck restrictions on 12[th],
and mechanical mixing from surface roughness (i.e. winds carrying TRAPs emitted on 12[th] being pushed up over the densely
spaced buildings between the roadway and monitor, resulting in diluted or no TRAPs measured at ground-level). Contrasting
this upwind definition with measurements from the sector 315°-345° in Fig. 3, which includes the major roadway Broadway
250 m from the receptor (farther than 12[th]), there is a difference in average $CO_2$ concentrations of about 15 ppm, and this
difference is likely due to reduced surface roughness NNW of the receptor. Both NR-TOR-2 and NR-VAN provide examples
of the complexity of siting near-road stations, and how site-specific considerations must be made when associating data with
meteorology.
**3.3 Time-series analysis**
Extracting information from one-dimensional ambient pollution time-series data (i.e. concentration as a function of time) for
the purpose of source apportionment is appealing as it allows the possibility of obtaining local and background estimates
without the need for more rigorous chemical analysis, computationally expensive multivariate analyses, or measurements made
at multiple locations. Many of such algorithms make use of the underlying principle that signal frequency is inversely related
to source distance. Regional or background sources (farther away from a receptor) produce slower varying, lower frequency
signals, whereas local (nearby) sources, such as traffic, produce faster varying, higher frequency signals (Tchepel and Borrego,

245    2010).





The frequency at which data is acquired limits the highest frequencies separable by such a method. Daily averages, for example,
are too lengthy to capture processes whose time scales are much shorter—a plume from a nearby on-road vehicle, for example,
would have a characteristic time on the order of seconds to minutes. Therefore, in order to isolate these local temporal
fluctuations, relatively high time resolution data are necessary. Sabaliauskas et al. (2014) made use of the wavelet
decomposition algorithm applied to one-minute particle number concentrations in an urban environment—a technique
described originally by Klems et al. (2010) and used historically in signal denoising and compression—in order to obtain local
and background UFP concentrations as a function of time. A similar technique was pioneered more recently by Wang et al.
(2018) in order to determine above-background pollutant concentrations for use in the determination of fleet-averaged emission
factors, and it is this technique that is explored further in this paper.

### 3.3.1 Interpolation of windowed minima

The time-series analysis algorithm explored in this paper is an interpolation of minimum values across a variable time window,
the duration of which effectively defines, in a sense, a cut-off frequency for local and urban background signal differentiation.
An average of linear interpolations taken across variable time windows was used here, as developed, validated, and utilized
by Wang et al. (2018), and is described in full detail therein along with code compatible with IGOR Pro 6.37. This algorithm
is henceforth referred to as method 3. This method yielded a baseline function, $C_B$, based on input near-road concentrations,
$C_{NR}$, constrained to yield non-negative solutions for each observation, i. Average local concentrations from method 3, $C_{L,3}$,
were then calculated using Eq. (4) and Eq. (5):
$$C_{L,3}[i] = C_{NR}[i] - C_B[i], \qquad C_B \leq C_{NR} \forall t \ , \tag{4}$$
$$\bar{C}_{L,3} = \frac{1}{N} \sum_{i=1}^{N} C_{L,3}[i] \ , \tag{5}$$
Again, $C_B$ are background concentrations determined algorithmically, and are a function of $C_{NR}$, whereas $C_{BG}$, as in Sect. 3.1,
are physically measured concentrations. It is worth noting that while the constraint $C_B \leq C_{NR} \forall t$ was applied in this algorithm,
it is not always the case that a background station will measure less than a near-road station during a given hour for a number
of different reasons. For example, Sofowote et al. (2018) showed that a receptor 167 m from the edge of Highway 401 measured
PM$_{2.5}$ concentrations that exceeded concurrent measurements at NR-TOR-1 (10 m from the edge of the highway) ~5% of the
time based on half-hourly measurements. Regardless, the impact of this assumption on estimated average local concentration
is likely minimal. In using this algorithm, the width of the averaging window will affect the resulting baseline—windows that
are shorter in duration will result in more temporally varying baselines, while longer windows will result in flatter baselines.
For information regarding function input parameters please refer to Wang et al. (2018)—equivalent parameters were chosen
in this study.



## 4 Results

### 4.1 Average differences between near-road and background sites

Over the duration of the study period average $C_{L,1}$ values were calculated using method 1, as described in Sect. 3.1, with resulting differences summarized in Table 2. Note that no $CO_2$ difference was calculated between Vancouver stations because $CO_2$ was not measured at BG-VAN.

The background-subtracted differences were smallest at NR-TOR-2; for every TRAP measured, both NR-TOR-1 and NR-VAN saw greater $C_{L,1}$ concentrations in comparison. This pattern is consistent with the lower traffic volumes at NR-TOR-2. Surprisingly, despite the drastic difference in traffic intensities between NR-VAN and NR-TOR-1, $C_{L,1}$ values at both sites were remarkably similar for most TRAPs. This similarity was in part due to NR-VAN's closer proximity to the roadway (6 m) compared with NR-TOR-1 (10 m), in conjunction with the significant fraction of diesel vehicles passing along Clark Drive (Wang et al., 2018). While most $C_{L,1}$ concentrations were similar between these two locations, UFPs at NR-TOR-1 were significantly greater ($2.8E+4$ vs. $1.5E+4$ $cm^{-3}$). However, this may be due to seasonal bias in UFP data availability (Table S1) between NR-TOR-1 and BG-TOR-1 (note especially the lack of concurrent data during summer months when ambient UFP concentrations are often lowest).

The $NO/NO_2$ ratios for $C_{L,1}$ at NR-TOR-2 were also markedly lower than the other near-road sites; these ratios at NR-VAN, NR-TOR-1, and NR-TOR-2 were, on average, 4.5, 2.0, and 0.7, respectively. A potential explanation for this is the relative residence times of vehicle plumes prior to detection at each site: because NR-VAN was positioned closest to the roadway, it is likely that vehicle plumes were fresher upon detection, whereas NR-TOR-2 sampled within an urban canyon where air tends to stagnate and recirculate. These results emphasize an important implication for near-road monitoring policies: while $NO_2$ alone is often regulated because of associated health effects, measurements of only $NO_2$ may not be a reliable metric for assessing near-road health impacts, as characteristics of the site may result in largely varying $NO/NO_2$ ratios.

The average differences for $O_3$ were negative, indicating that ozone concentrations tend to be lower near major roads. Ozone is presumably being titrated due to the higher near-road concentrations of NO. Furthermore, $O_3$ production in downtown Toronto and metropolitan Vancouver generally occurs in a VOC-limited regime, meaning that the additional $NO_x$ near roads does not enhance local ozone formation (Ainslie et al., 2013; Geddes et al., 2009).

While $PM_{2.5}$ is generally considered to be a more regional and homogenous pollutant in urban environments, the observed values of $C_{L,1}$ (1.48, 0.27, and 2.26 $\mu g$ $m^{-3}$ at NR-TOR-1, NR-TOR-2, and NR-VAN, respectively) were found to be significantly greater than zero, and may be indicative of both primary tailpipe and non-tailpipe (e.g. brake wear, road dust resuspension, etc.) emissions. A recent study by Jeong et al. (2019) characterized the sources and composition of $PM_{2.5}$ at both NR-TOR-1 and NR-TOR-2 using an X-ray fluorescence continuous metals monitor. They found that while concentrations of aged organic aerosol, sulfate, and nitrate were similar between the two sites, contributions from sources such as traffic exhaust, brake wear, and road dust differed significantly, and were the primary factors responsible for differences in average $PM_{2.5}$



concentrations. Another study by Sofowote et al. (2018), examined in more detail the reasons for elevated PM$_{2.5}$ constituents
at NR-TOR-1, with particular emphasis on BC, relative to another receptor 167 m from Highway 401.

## 4.2 Downwind-upwind pollutant differences

As stated previously, NR-TOR-1 was the most ideal near-road monitoring location in this study for associating TRAP
measurements with local meteorology, as it was positioned on flat terrain, and the major roadway which it was stationed next
to was the only significant source of TRAPs in the immediate area. Thus, the direction of wind at this site had a significant
impact on measured pollutant concentrations (Fig. 1). Using the methods described in Sect. 3.2, hourly TRAP concentrations
were aggregated based on wind direction, and were classified as either being downwind, upwind, or neither. Downwind and
upwind averages were calculated across the entirety of the study period and their differences, C$_{L,2}$ are summarized in Table 3
for all near-road sites. Additional information regarding the number of downwind/upwind hours and confidence intervals are
provided in the supplementary information (Sect. S2).
The C$_{L,2}$ values reported in Table 3 for NR-TOR-1 correspond relatively well with, but are higher than, the C$_{L,1}$ values in Table
2. This is true for most pollutants, with the exception of O$_3$ and PM$_{2.5}$. The reason these C$_{L,2}$ values are generally greater than
their corresponding C$_{L,1}$ values is believed to be due to the following reason: when a site is directly downwind from a road it
will generally experience greatest TRAP concentrations, as it is this case in which there is the smallest distance for dilution
between the road and the site.  In contrast, the C$_{L,1}$ values reported in Table 2 were averaged across all meteorological scenarios.
The fundamental difference between methods 1 and 2 is explored further in Sect. S3 in the supplementary information.
Unlike NR-TOR-1, NR-TOR-2 was not an ideal site for applying method 2 in a straightforward manner, as it measured air
samples within an urban canyon where micrometeorology was complicated by vortices, stagnation, and recirculation effects.
Using the downwind and upwind sector definitions in Sect. 3.2.2, C$_{L,2}$ values were calculated at NR-TOR-2 and are
summarized in Table 3. This methodology of contrasting downwind and upwind pollutant averages at NR-TOR-2 was unable
to produce meaningful differences and the resulting disagreement with the near-road-urban-background differences in Table
2 is evident. Associating ground-level TRAP concentrations with city-scale meteorology at this site was complicated by
surrounding urban architecture and the presence of an intersection approximately 50 m SW of the receptor. In actuality, the
difference calculated for this site was between that of leeward and windward in-canyon concentrations, and this difference was
not as substantial as the NR-TOR-2 and BG-TOR-2 average site difference. For these reasons, associating near-road pollutant
concentrations with meteorological data was not an effective way of differentiating between local and regional influences on
pollutant concentrations at this particular near-road site. In general, in order to attain this differentiation for measurements
made in urban canyons, more complicated meteorological models are necessary; hence, simple downwind/upwind differences
are not universally applicable to near-road monitoring data, especially for locations in heavily urbanized landscapes.
Lastly, the siting of NR-VAN was somewhere between NR-TOR-1 and NR-TOR-2 in terms of complexity in associating
TRAP concentrations with meteorology. The presence of densely spaced residential buildings within the immediate vicinity
of the measurement station resulted in surface roughness having an effect on winds carrying TRAPs from major roadways





farther away. Despite this, the differences between average downwind and upwind TRAP concentrations at NR-VAN were
similar to, albeit larger, than the NR-VAN/BG-VAN differences in Table 2, a result similar to that for NR-TOR-1. The fact
that consistent results were seen for NR-VAN and NR-TOR-1 but not NR-TOR-2 underlines the importance of a station's
location, surrounding obstructions to winds, and location of traffic sources, and that associating near-road TRAP
concentrations with meteorological variability should be done with caution, taking into account the subtleties of each site's
environ. The apparent stronger influence of the intersection rather than traffic directly next to NR-VAN (i.e. winds originating
from 90°; see Fig. 3), despite Clark Drive being 6 m vs the intersection being 65 m away, may seem paradoxical. We speculate
that the acceleration of southbound traffic along Clark Drive at this intersection was the main source of emissions, while
coasting past the site, particularly when slowing down for the stop light, would have contributed much less.

**4.3 Local concentrations determined using time-series analysis**

Method 3, as described in Sect. 3.3.1, was applied to hourly pollutant concentrations, and the algorithm input parameters were
chosen based on those previously validated by Wang et al. (2018) at the same near-road locations. From the output, $C_{L,3}$ was
determined as a function of time, and then averaged across the entirety of the measurement campaign; the resultant averages
are summarized in Table 4.
A key benefit to this method was that it was able to estimate local and background $CO_2$ concentrations at NR-VAN, where
$CO_2$ measurements were made only in the near-road environment and not at the background site. This emphasizes a key
advantage to approaches such as these: traffic-related signal can be isolated from near-road measurements alone, without the
need for background or even meteorological measurements. Furthermore, this differentiation was performed on an hourly
basis, thereby retaining information in the time domain, which was not possible with method 2.
Across all near-road locations, average $C_{L,3}$ concentrations were quite similar to respective average $C_{L,1}$ values, implying that
method 3, which uses only near-road data, is a robust means of estimating urban background and local traffic-related pollutant
concentrations. This was true even for NR-TOR-2, where micrometeorology complicated analysis using method 2. Pollutants
that are exceptions to this are $O_3$ and $PM_{2.5}$. Because $O_3$ is formed through secondary processes, and because it is often inversely
correlated with primary traffic emissions, it is not sensible to attempt to attribute its ambient concentrations to local or
background sources using method 3, and so results for this pollutant are omitted in Table 4. As for $PM_{2.5}$, because its signal
was largely dominated by regional-scale sources and dynamics, temporal fluctuations in roadside $PM_{2.5}$ concentrations
generally varied more slowly than those of primary pollutants such as NO or BC, for example. Furthermore, this variability is
generally meteorologically-driven and occurs homogeneously over large areas (10s of kilometres); we posit that these
variabilities associated with meteorology were falsely attributed to local signal, causing local $PM_{2.5}$ concentrations ascertained
through this method to be much higher than those in Table 2.
Although application of method 3 is not suitable for some pollutants (i.e. regional, secondary pollutants such as $O_3$ and $PM_{2.5}$),
it appears to behave in an accurate and robust manner for most others. Comparing local concentrations in Table 4 with those



listed in Table 2, it appears that method 3 produces similar results when compared with method 1, with the added benefit of
retaining information in the time domain and not requiring a second site.
**4.4 Comparison of background subtraction methods**
Three techniques were applied to the near-road monitoring locations in this study to extract information regarding local TRAP
concentrations: 1. Average differences between near-road and urban background locations, 2. Downwind-upwind differences
in near-road measurements, and 3. Average concentrations inferred through time-series analysis of near-road data. Generally,
methods 1 and 3 agreed well with one another, whereas method 2 produced values that were high in comparison with the other
two methods at NR-TOR-1 and NR-VAN, and generated results that were close to zero at NR-TOR-2. A comparison of the
three methodologies is summarized graphically in the supplementary information (Fig. S1-S3). The close agreement of
methods 1 and 3, which describe the average concentrations attributed to local traffic, is encouraging, suggesting that both
methods are applicable for the estimation of local traffic impacts on ambient air quality at near-road stations. The consistently
higher values for method 2 highlight the drawbacks of relying exclusively on wind direction data for source apportionment
efforts.
**4.5 Application of local concentrations**
Subtraction of background concentrations allows the influences of local traffic on near road TRAP concentrations to be
assessed. The benefits in terms of improved understanding were examined and illustrated by applying the local concentrations
thereby derived in two ways. The degree to which traffic influences TRAP concentrations beside a road can vary day-to-day
depending on the prevailing meteorology. Using the local signal allowed the magnitude of this source of variability to be
assessed in a manner that is consistent across all TRAPs and across all near-road sites. In contrast, the contribution of traffic
to the total concentration will differ across pollutants. For example, some pollutants such as NO may be predominantly from
traffic while others such as $CO_2$ will be dominated by the background. Separating the local and background concentrations
allowed assessment of how the portion from local traffic varied between sites and across the pollutants. Effectively, the
background subtraction methodology provided estimates that illustrate how much concentrations beside a road would drop if
all the traffic on that road were to be removed, as concentrations would converge to that of the urban background in that case.
**4.5.1 Effect of meteorology on local TRAP variability**
Using the hourly values of $C_{L,3}$ at each near-road station determined using method 3 in Sect. 3.3.1, the roles of individual
meteorological parameters on the variability of these local concentrations were explored. While roadside concentrations are
affected by meteorology in a number of ways, local pollutant quantities—of interest are those from vehicular exhaust—are
expected to behave in a more predictable manner in comparison, and indeed there are many means in which to predict the
evolution of these exhaust plumes, from simple dispersion models to computational fluid dynamics. Here, however, a more
simplified means of underlining the effect of wind on above-background TRAP concentrations was utilized: local TRAP



concentrations normalized to their mean values were associated with both the direction and speed of local winds, the former
showing the effect of downwind/upwind variability and the latter showing that of dilution. Normalization allowed results to
be generalized between sites and pollutants where mean emission rates of TRAPs may differ. Because NR-TOR-2 was situated
within an urban canyon, the effect of meteorology on its measured concentrations was not generalizable to the other two
stations in this study; for this reason it is omitted from this section.

**4.5.2 Wind direction**

Wind direction can have a large influence on roadside TRAP concentrations. Shown in Fig. 4 is the dependence of normalized
local pollutant concentrations on wind direction at both NR-VAN and NR-TOR-1. Generally, downwind measurements have
the effect of enhancing local concentrations by a factor of ~1.5-2.0, whereas upwind conditions suppress local concentrations
by a factor of ~4.0, with respect to the mean. Note that these upwind concentrations did not necessarily converge to zero as
hourly averages in which winds were from a given direction 75% of the time were utilized. It is also conceivable that during
upwind periods, local turbulence from traffic and/or brief shifts in wind direction resulted in some degree of plume capture. It
would appear that, on an hourly-averaged basis, traffic's contribution to local TRAP variability (i.e. irrespective of background
pollution) at a roadside receptor may change by a factor of six to eight depending on the average direction of wind.
As shown in Fig. 4, a clear sinusoidal wind direction dependency is apparent at NR-VAN and NR-TOR-1, with similar ranges
in enhancement and suppression at both sites. However, at NR-VAN, there appears to be two modes in concentration
enhancement. The Clark Drive and 12$^{th}$ Avenue intersection, located approximately 65 m from the receptor, had an influence
on local TRAPs originating from the south. However, given its distance, west/eastbound traffic along 12$^{th}$ avenue should not
have had an influence similar to that of Clark Drive which was only 6 m away. We postulate that the traffic lights at the
intersection caused stop-and-go patterns in which southbound traffic on Clark Drive was often backed up to the monitoring
location, and it is these driving patterns that are believed to be associated with the enhancement seen between the wind
directions of 100°-200° at NR-VAN.
When comparing methods of background subtraction, it was shown that method 2 yielded higher estimates of the local
concentrations in comparison with the other two methodologies, as further explored in Sect. S3 of the supplementary data.
Across pollutants, it was found that on average this downwind/upwind difference resulted in local TRAP concentrations that
were factors of 1.3 and 1.4 times greater than those inferred from method 3 at NR-VAN and NR-TOR-1, respectively (Table
S6). In short, this corresponds well with above-average normalized local pollutant concentrations during downwind conditions
at both sites (Fig. 4), during which conditions values of $C_{L,3}$ were found to be similar factors greater than the mean at both sites
(Table S6).
Lastly, it is of interest to note that hourly upwind $C_{L,3}$ concentrations at either site yielded non-zero local concentrations. It is
indeed likely that at an hourly time-resolution some plume capture will occur during predominately upwind conditions;
however, this seems to carry with it the implication that upwind analysis at a near-road location may overestimate background
concentrations. To test this, average upwind concentrations were compared with average concentrations measured at each




nearest background location, the results of which are summarized in Table S5. Generally, the two appear to agree well with
one another, and so any plume capture during upwind conditions likely produced a negligible impact on total concentrations.

**4.5.3 Wind speed**

Similar to the analysis in the previous section, the effect of wind speed on roadside TRAP concentrations was explored at NR-
TOR-1 and NR-VAN, and consistent results were found between them. Under stagnant conditions (wind speeds of ~1.0 m s$^{-}$
$^{1}$), local pollutant quantities were found to be enhanced by factors of ~2.0 and ~1.7 at NR-VAN and NR-TOR-1, respectively,
and high wind speeds (> 10 m s$^{-1}$) suppressed these quantities by a factor of ~2.0 at both sites (Fig. 5), giving an overall
influence factor of 3.4 to 4. The maximum levels of enhancement and suppression were slightly smaller than the results found
for wind direction, implying a slightly smaller or equivalent importance on local TRAP concentrations at a given roadside
receptor. The relation used to model the effect of wind speed on normalized local concentrations was the following:
$$\frac{C_{L,3}}{\bar{C}_{L,3}} = \frac{c_1}{WS^{c_2}},$$    (6)
where $C_{L,3}$ are local pollutant concentrations determined through method 3, $c_1$ and $c_2$ are regression parameters, and WS is
wind speed as measured at the station. Indeed, more involved models have been shown to better represent the wind speed
dependency of specific pollutants (Jones et al., 2010); however, simplicity is preferred here so as to generalize results across
sites and pollutants.
On average, the regression parameters $c_1$ and $c_2$ were found to be ~2.0 and ~0.6 for NR-VAN, and ~1.6 and ~0.5 for NR-TOR-
1, respectively (Table S7). While different $c_1$ parameters were determined for both sites, presumably due to their difference in
roadway proximity, similar $c_2$ parameters between 0.5-0.6 were found. The $c_2$ parameter, which embodies the wind speed-
pollutant decay relationship, is expected to be independent of a station's proximity to the roadway. As with the wind direction
analysis in the previous section, these associations with respect to wind speed were averaged from two years of hourly data
across the entire study domain, meaning they were acquired from a range of pollutants, traffic conditions, wind directions, and
times of day. While less descriptive from a mechanistic perspective, these results are intended to be more representative of the
ranges of variability in average above-background exposure levels in the immediate area.

**4.6 Fraction of near-road pollution attributable to local sources**

The time-series based estimates of the background concentrations were also applied to estimate the portion of the pollutant
concentrations that were due to local traffic. For example approximately half of total BC concentrations were estimated to be
due to local sources at NR-TOR-1 with lower and higher percent contributions at NR-TOR-2 and NR-VAN, respectively (Fig.
6).  The contribution of local sources varied across the pollutants; NO had the highest local contribution at the near road sites
while $CO_2$ had the lowest (Fig. 7). Further, this methodology was able to replicate trends in weekday/weekend background
pollution variability—shown in Fig. 6 is BC, for example, with others in the supplementary (Fig. S4-S9). Local components



of air pollution showed far greater differences between weekdays and weekends at each near-road monitoring location,
emphasizing the effect of different on-road traffic conditions between these two sets of days. Generally, TRAP concentrations
measured at urban background sites were slightly higher on weekdays compared to weekends, and this change in regional
pollution was captured in the background contributions extracted from the near-road data. It should be expected that average
concentrations measured at BG-TOR-1 should match the background elements of NR-TOR-1 reasonably well, with a similar
argument to be made for BG-TOR-2 and NR-TOR-2; however, these urban background concentrations are likely not perfectly
homogeneous throughout the city. The spatial difference between BG-TOR-1 in north Toronto and BG-TOR-2 in south
Toronto was 20 km, and the difference in average pollutant levels between the two reflects this.

## 5 Conclusions

In this study TRAP concentrations were measured continuously at time resolutions of one hour or finer for over two years at
three near-road and three urban background locations. Three methods were explored for estimating the contribution of local
and regional/background sources on near-road measurements: differences between average measurements taken near-road and
at a nearby urban background location, downwind-upwind analysis at the near-road location, and time-series analysis of near-
road pollutant data. Generally, the near-road vs urban background and time-series analysis methods produced results that were
in good agreement; these values represent contributions to TRAP due to local traffic averaged over all wind directions. The
downwind-upwind method yielded local concentrations that were higher than the average station differences by approximately
40%; this was attributable to the downwind/upwind analysis isolating the conditions where traffic has the greatest impact on
a site while the average differences included data across all wind conditions.
The time-series analysis method was an accurate and robust means of differentiating local and regional signal, with the added
benefits of being applicable across all near-road sites, not being constrained to certain meteorological scenarios or requiring a
separate background site, and retaining information in the time domain. This methodology is recommended for future use in
applications such as: determining the impact of local on-road traffic to a roadside receptor, isolating background concentrations
from ambient data for use in dispersion modelling, and obtaining above-background concentrations for fleet emission factor
calculations.
Lastly, to demonstrate the value in isolating the influence of local sources at an hourly time resolution, local TRAP
concentrations determined using time-series analysis were compared with meteorological variables at two of the near-road
sites, NR-VAN and NR-TOR-1. This analysis yielded trends that were generalizable across all measured pollutants, with the
exception of $PM_{2.5}$ and $O_3$. Wind direction had a factor of influence of approximately seven at both near-road sites, while the
effect of wind speed was found to be slightly smaller, varying local hourly concentrations by a factor of four, with highest
concentrations seen during stagnant conditions and lowest concentrations as wind speed became large. Both sites exhibited
similar decays in local concentration with respect to wind speed; proportionality to wind speed was found to be between $WS^{-0.5}$
and $WS^{-0.6}$.



**Author contribution**

AM, LW, CA, DH, JRB, and GJE designed and initiated the near-road monitoring study. Data collection and quality assurance
from Torontonian stations was performed by: NH, JMW, CHJ, RMH, US, JD, YS, and MN, while GD was responsible for the
two stations in Vancouver. NH prepared the manuscript, with contributions from all co-authors, and performed all data
analysis.

**Acknowledgements**

We would like to thank all partners involved in the near-road monitoring pilot project in Canada, including staff from Metro
Vancouver, the Ontario Ministry of the Environment Conservation and Parks, and Environment and Climate Change Canada
for their assistance in formulating the design of the study, as well as, deploying and maintaining the air quality instruments
used in this study.

**Competing interests**

The authors declare they have no conflict of interest.




















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





**Table 1: IDs, locations, name of major roadway, and average daily traffic intensity for each monitoring location.**

| Station ID | Latitude | Longitude | Major Roadway | Annual Average Daily Traffic (AADT) | Distance from Roadway [m] |
|---|---|---|---|---|---|
| NR-TOR-1 | 43.7111 | -79.5433 | Highway 401 | 405,500 | 10 |
| BG-TOR-1 | 43.7806 | -79.4675 | - | - | - |
| NR-TOR-2 | 43.6590 | -79.3954 | College Street | 17,200 | 15 |
| BG-TOR-2 | 43.6122 | -79.3887 | - | - | - |
| NR-VAN | 49.2603 | -123.0778 | Clark Drive | 33,100 | 6 |
| BG-VAN | 49.2529 | -123.0492 | - | - | - |



**Table 2: Local pollutant concentrations determined using method 1 for each near-road and urban background station pair ($C_{L,1}$).**
**Number of coincidental hours, N, and mean values with their respective 95% confidence intervals are also reported.**

| Pollutant | $C_{L,1}$ TOR-1 | | $C_{L,1}$ TOR-2 | | $C_{L,1}$ VAN | |
|---|---|---|---|---|---|---|
| | N | $\mu \pm$ 95% CI | N | $\mu \pm$ 95% CI | N | $\mu \pm$ 95% CI |
| NO [ppb] | 14169 | 21.5 ± 0.4 | 13768 | 3.5 ± 0.1 | 10647 | 23.0 ± 0.5 |
| $NO_2$ [ppb] | 13765 | 8.7 ± 0.1 | 11211 | 5.4 ± 0.1 | 10666 | 5.1 ± 0.1 |
| CO [ppb] | 6479 | 103.2 ± 2.7 | 13603 | 72.3 ± 1.5 | 9435 | 95.7 ± 2.3 |
| $CO_2$ [ppm] | 7900 | 14.4 ± 0.6 | 10686 | 10.6 ± 0.4 | - | - |
| $O_3$ [ppb] | 13753 | -5.9 ± 0.1 | 15109 | -2.9 ± 0.1 | 10535 | -3.9 ± 0.1 |
| $PM_{2.5}$ [$\mu g\ m^{-3}$] | 14170 | 1.48 ± 0.06 | 15193 | 0.27 ± 0.05 | 10491 | 2.26 ± 0.07 |
| UFP [$cm^{-3}$] | 5212 | 29600 ± 800 | 7400 | 7400 ± 200 | 9452 | 11600 ± 300 |
| BC [$\mu g\ m^{-3}$] | 8036 | 1.03 ± 0.03 | 14740 | 0.34 ± 0.01 | 10728 | 1.18 ± 0.02 |


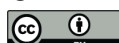



**Table 3: Pollutant averages aggregated by downwind, $C_{DW}$, and upwind, $C_{UW}$, conditions at each near-road site, along with the**
**respective differences, $C_{L,2}$.**

| Pollutant | NR-TOR-1 | | | NR-TOR-2 | | | NR-VAN | | |
|---|---|---|---|---|---|---|---|---|---|
| | $C_{DW}$ | $C_{UW}$ | $C_{L,2}$ | $C_{DW}$ | $C_{UW}$ | $C_{L,2}$ | $C_{DW}$ | $C_{UW}$ | $C_{L,2}$ |
| NO [ppb] | 37.8 | 2.9 | 34.9 | 6.0 | 3.2 | 2.8 | 56.6 | 9.7 | 46.8 |
| $NO_2$ [ppb] | 21.2 | 10.7 | 10.5 | 8.5 | 10.4 | -1.9 | 21.9 | 11.5 | 10.4 |
| CO [ppb] | 364.4 | 226.6 | 137.9 | 247.9 | 246.8 | 1.1 | 414.3 | 210.1 | 204.2 |
| $CO_2$ [ppm] | 437.3 | 416.4 | 20.9 | 423.1 | 421.4 | 1.7 | 461.6 | 414.5 | 47.1 |
| $O_3$ [ppb] | 15.3 | 33.2 | -17.9 | 24.2 | 28.7 | -4.5 | 9.4 | 19.7 | -10.3 |
| $PM_{2.5}$ [µg m$^{-3}$] | 7.68 | 9.01 | -1.33 | 3.80 | 9.01 | -5.21 | 8.81 | 5.57 | 3.23 |
| UFP [cm$^{-3}$] | 57000 | 15300 | 41700 | 12900 | 16700 | -3800 | 30000 | 14000 | 16000 |
| BC [µg m$^{-3}$] | 2.13 | 0.73 | 1.40 | 0.63 | 0.81 | -0.18 | 2.48 | 0.84 | 1.64 |




**Table 4: Average TRAP concentrations associated with local influences at each near-road monitoring location calculated using method 3, $C_{L,3}$, along with number of hours (N) and 95% confidence intervals (CI) on the means ($\mu$).**

| Pollutant | $C_{L,3}$ TOR-1 | | $C_{L,3}$ TOR-2 | | $C_{L,3}$ VAN | |
|---|---|---|---|---|---|---|
| | N | $\mu \pm$ 95% CI | N | $\mu \pm$ 95% CI | N | $\mu \pm$ 95% CI |
| NO [ppb] | 15524 | $18.3 \pm 0.4$ | 14937 | $3.8 \pm 0.1$ | 15134 | $27.6 \pm 0.6$ |
| $NO_2$ [ppb] | 15087 | $9.2 \pm 0.1$ | 12359 | $5.3 \pm 0.1$ | 15148 | $9.7 \pm 0.1$ |
| CO [ppb] | 13008 | $114.6 \pm 2.2$ | 15152 | $68.7 \pm 1.3$ | 13935 | $153.3 \pm 3.4$ |
| $CO_2$ [ppm] | 14812 | $19.6 \pm 0.4$ | 14626 | $13.3 \pm 0.2$ | 13503 | $39.0 \pm 0.7$ |
| $PM_{2.5}$ [$\mu$g m$^{-3}$] | 15484 | $4.30 \pm 0.08$ | 15730 | $2.92 \pm 0.06$ | 14879 | $3.99 \pm 0.10$ |
| UFP [cm$^{-3}$] | 12683 | $22754 \pm 449$ | 14931 | $7088 \pm 108$ | 14463 | $15252 \pm 251$ |
| BC [$\mu$g m$^{-3}$] | 15443 | $1.01 \pm 0.02$ | 15451 | $0.41 \pm 0.01$ | 15312 | $1.26 \pm 0.02$ |



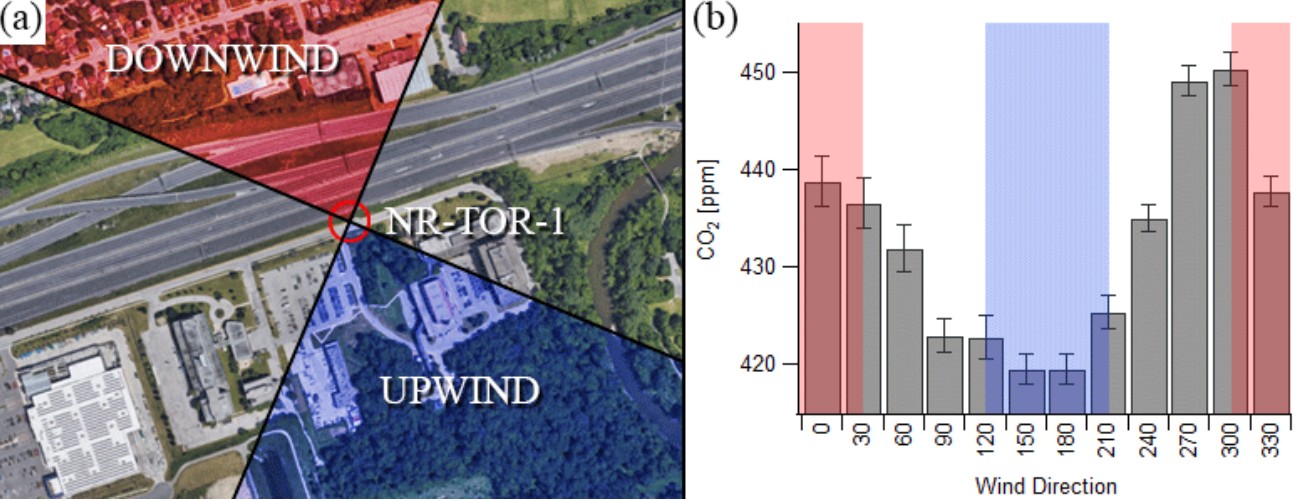

**Figure 1: Satellite image of the NR-TOR-1 site, along with upwind (blue) and downwind (red) quadrant definitions. Meteorological measurements were taken on top of a 10 m mast at the location of the station (labelled: NR-TOR-1) (a). Average ambient $CO_2$ concentrations by wind direction, with upwind and downwind definitions again highlighted in blue and red, respectively. Error bars are 95% confidence intervals on the mean (b).**

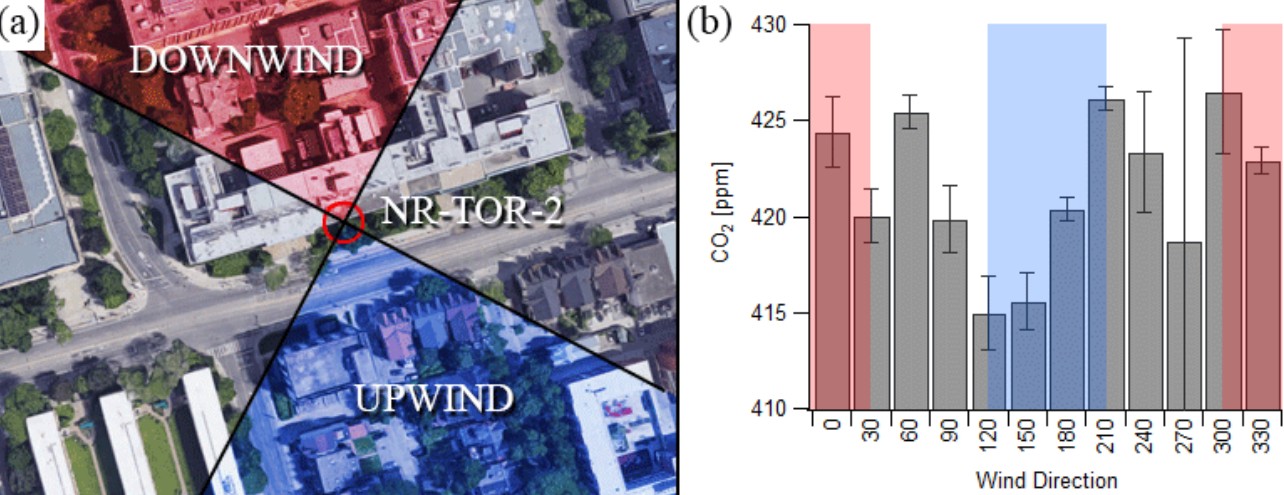

**Figure 2: Satellite image of the NR-TOR-2 site, along with upwind (blue) and downwind (red) quadrant definitions. Meteorological measurements were recorded on the roof of the facility (labelled: NR-TOR-2) (a). Average ambient $CO_2$ concentrations by wind direction, with upwind and downwind definitions again highlighted in blue and red, respectively. Error bars are 95% confidence intervals on the mean (b).**



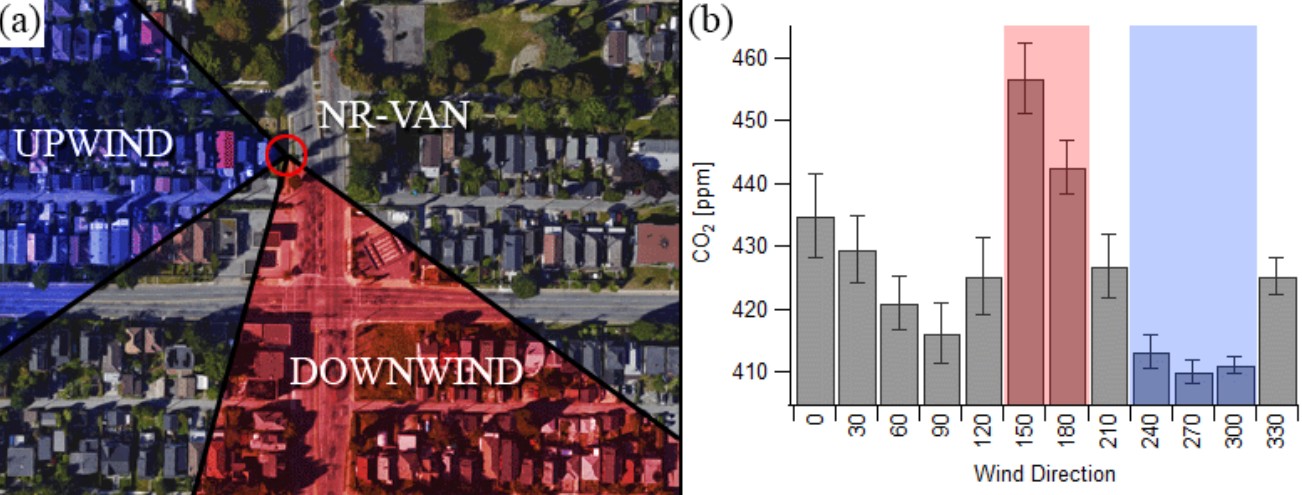

**Figure 3: Satellite image of the NR-VAN site, along with upwind (blue) and downwind (red) sector definitions. Meteorological measurements were recorded on a 10 m mast above the station's location (labelled: NR-VAN) (a). Average ambient $CO_2$ concentrations by wind direction, with upwind and downwind definitions again highlighted in blue and red, respectively. Error bars are 95% confidence intervals on the mean (b)**



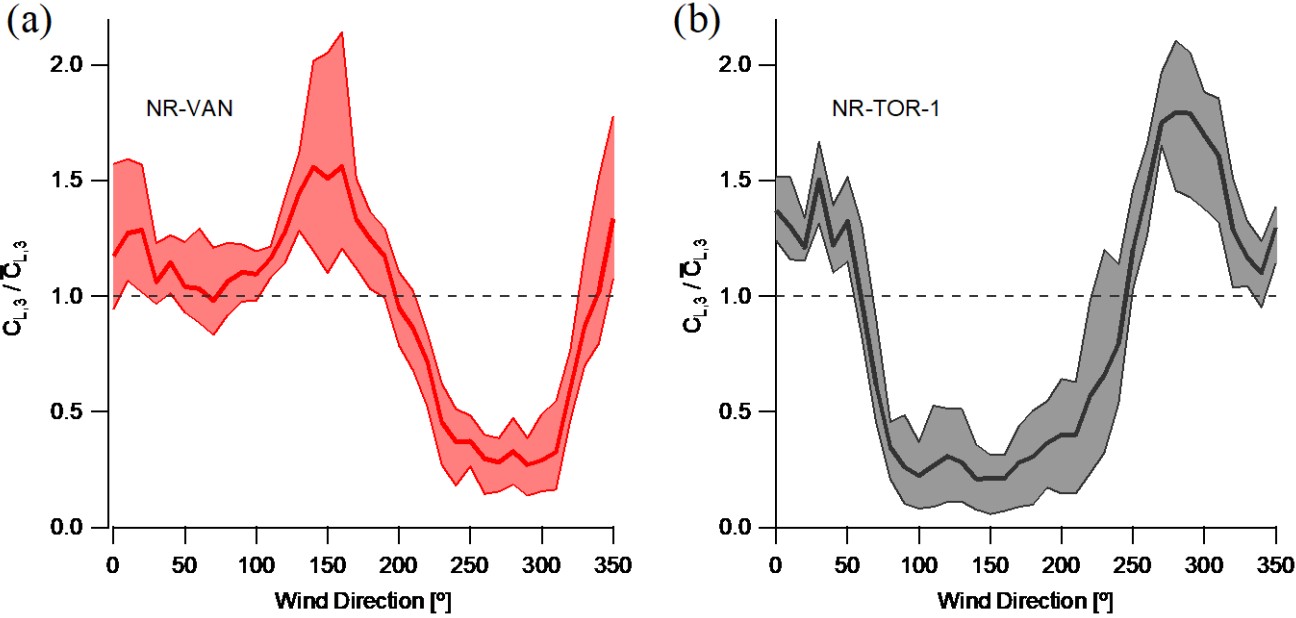


**Figure 4: Normalized local pollutant concentrations determined using method 3 as a function of wind direction at NR-VAN (a) and**
**NR-TOR-1 (b). Solid lines indicate the average trend amongst all TRAPs, and shaded areas indicate the range of variability between**
**TRAPs.**




















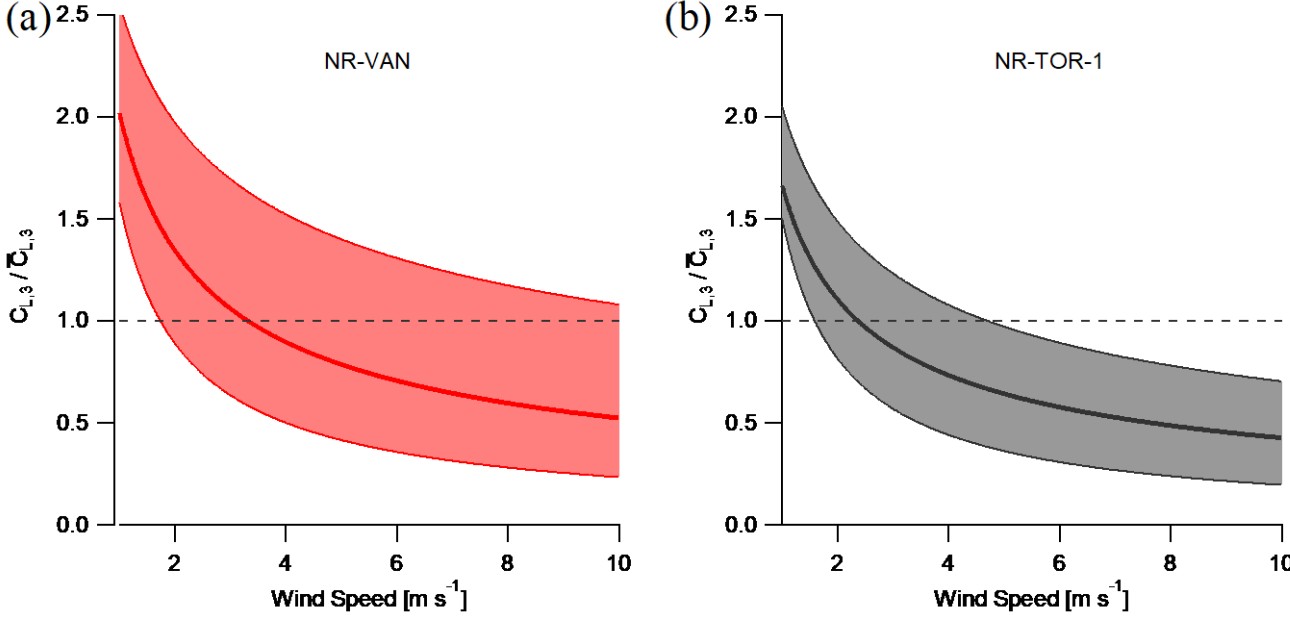


**Figure 5: Normalized local pollutant concentrations determined using method 3 as a function of wind speed at NR-VAN (a) and NR-TOR-1 (b). Solid lines indicate the average trend amongst all TRAPs, and shaded areas indicate the range of variability between TRAPs.**


















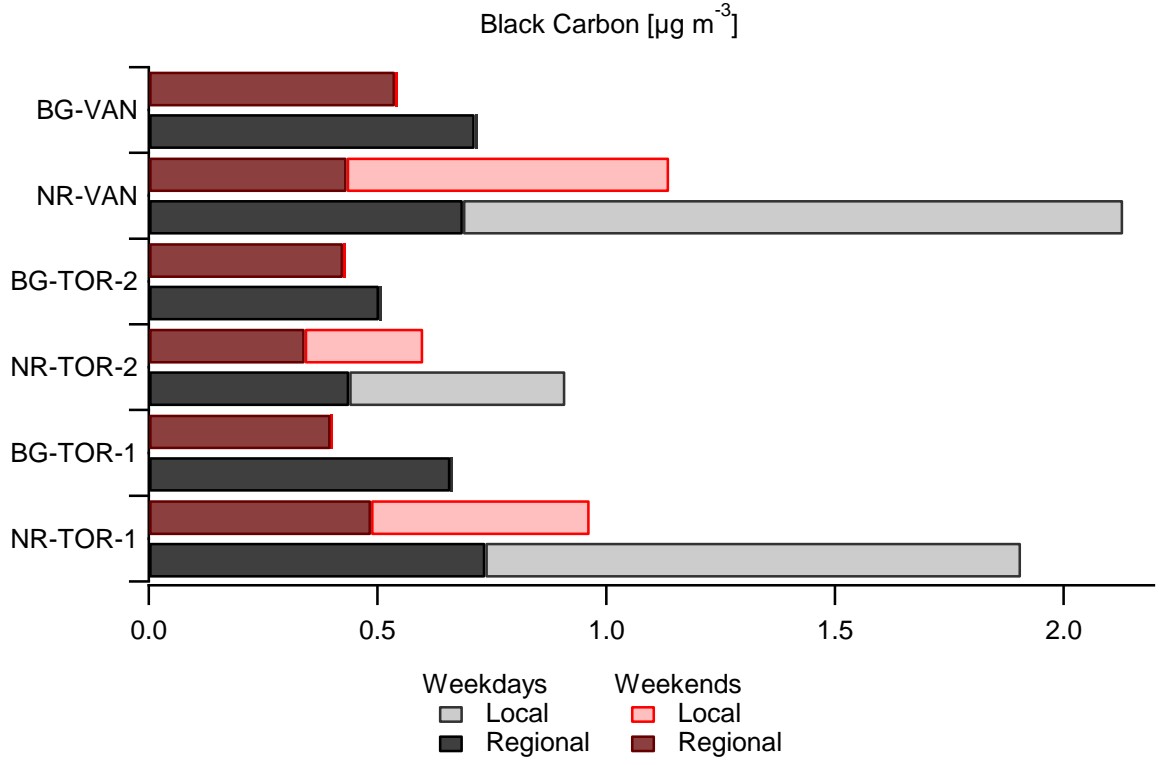


**Figure 6: Black carbon concentrations at each monitoring location in this study. Each site is separated by weekday and weekend, and bars are stacked according to concentrations attributed to local and regional sources. Background stations are presumed fully regional and therefore contain no local component.**




















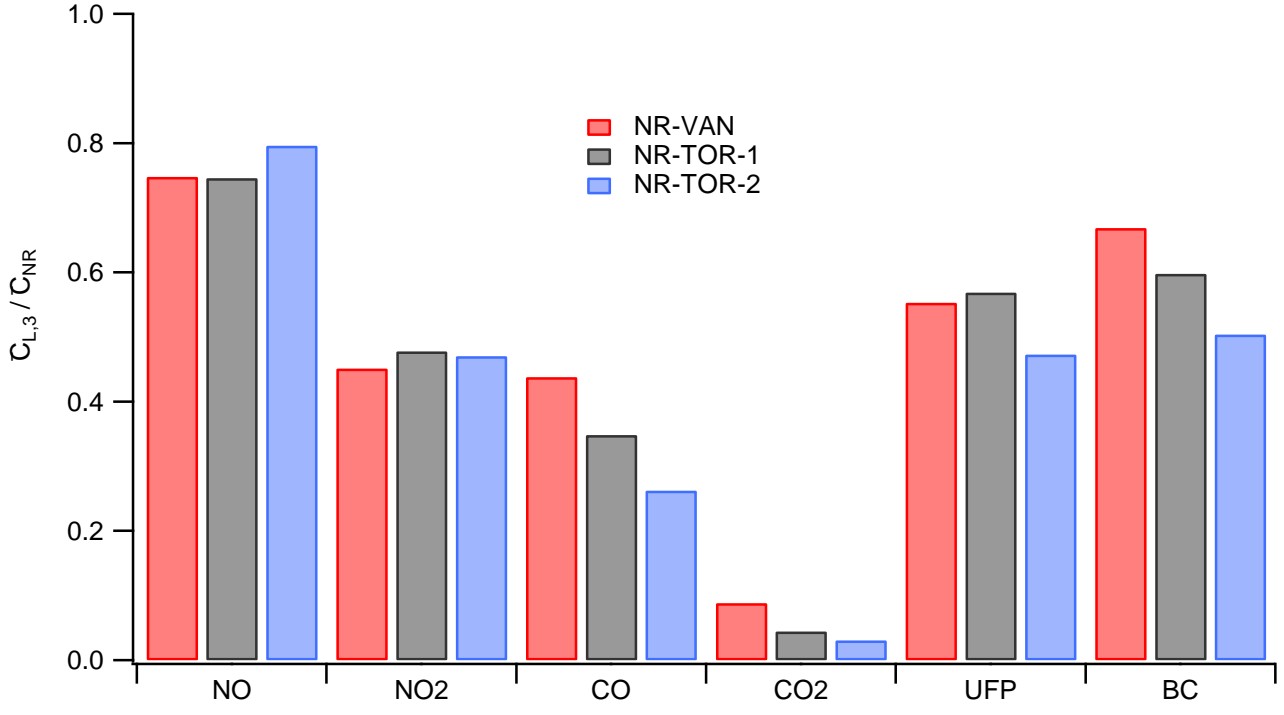


**Figure 7: Average fraction of near-road measurements attributed to local sources, as determined by method 3, for each near-road monitoring location.**