# Peer review of "Traffic-related air pollution near roadways: discerning local impacts"

_Atmospheric Measurement Techniques, 2019_

## Referee Comment (RC1) · Anonymous Referee #1 · 14 Apr 2019

This is a thorough application of three methods to distinguish traffic-related air pollutants from background concentrations, applied to three road-side stations in two cities in Canada. The results are novel, interesting, well presented. The manuscript should be accepted for AMT after authors' response to the issues as raised below.

Of course, it makes a big difference if a roadside station is located 6 m away from the nearest traffic line or 10 m. Actually, any different position may lead to different concentrations, because the variability of air pollutant concentrations varies largely spatially. The authors are aware of that. However, the tone sometimes suggests that the results (concentrations!) are transferable to other locations or situations. For example, the last sentence of the abstract ("Downwind conditions enhanced local concentrations by a factor of ∼2 relative to their mean, while upwind conditions suppressed them by a

[Figure]

factor of ∼4") is, one the one hand, perfectly fine. On the other hand, there is no caveat saying: "This applies to this very specific situation, don't generalize!" Also, referring to lines 289 – 299 and Table 2, the absolute numbers of C_L are not comparable to each other between sites, even though drastic differences between the sites are apparent. The authors are asked to go through their manuscript and find more cautious wording in this respect.

In the eyes of this reviewer, the data set allows much more interesting analysis of emission factors, for example between NOx and CO2. How do NOx/CO2 ratios or UFP/CO2 ratios compare o the results of similar studies? Similar applies to CO. It is however acknowledged that this is outside the scope of this study.

Section 4.3 and Table 4: Why not did authors apply his method for ozone? When using maxima instead of minima for the time series analysis this should be no problem to do. The justification given in lines 362 – 364 is no convincing. The urban background concentration of O3 could have been quantified that way, and be compared with the respective results of methods 1 and 2.

Eq. 2 seems screwed. Probably, a parenthesis is missing on the right-hand side, opening before C_NR[i] and closing after C_BG[i]

Line 187: A justification should be given as of why M and N are typically not identical. It is a bit counterintuitive.

line 194: Why did you chose 75 % here? Likely, the results are more reliable if 100 % is used. See also line 413.

The PM2.5 results are puzzling indeed. It could be the precision and accuracy of the analyzers not being able to resolve the small differences in concentrations between stations and within time series. If so, the results are not statistically robust. This issue should be analyzed in more detail and be presented and discussed in the manuscript.

Again, a fine analysis!

---

## Referee Comment (RC2) · Anonymous Referee #2 · 30 Apr 2019

This study provides three methods to quantify the contribution of local traffic emission to near road measurements and explored the effects of wind speeds and directions on near road pollution. The MS is well written and contains rich and interesting results. However, I worry that the background determined in the paper using method3 may not be a real background, but a characterization of the environment around the road. The authors need to give a clearer view on what they think it means. Below are some specific comments need a response before acceptance.

Specific comments:

1) The first paragraph in Introduction about exposure is not that relevant to the rest of the paper, so would better focus on the topic of traffic-related pollutants close to the road.

[Figure]

2) In L174, why the calculated average difference is expected to converge the true average difference between sites? Are the differences between sites normally distributed? Is this convergence non-trivial and not simply a property of averages or central limit theory?

3) In L190, since the authors have realized the downwind and upwind scenarios may encompass different time frames and may influence the results, why not do some statistical tests? It seems important to the final outcomes.

4) In L238-274, the method3 was not explained properly.

Firstly, 'Time-series analysis' seems too general and may not be a good subtitle here and in the rest part of the MS. It often implies decomposition and forecasting.

Secondly, the authors talked about the frequency of signals very often in the first two paragraphs (L238-254) and allude to the wavelet decomposition algorithm used by Sabaliauskas (2014) as similar to their method. But I think this is not quite right and misleading. What I expected after the description is a frequency analysis, but method3 is approximately a 'moving minimum' baseline algorithm. As an example of signal processing and a spatial frequency domain in the road-environment can be seen in Xing and Brimblecombe (2019). Although wavelet analysis can also be used to exact baselines as shown by Liland et al. (2010), the underlying theory is different. There have been many baseline algorithms in Liland et al review (2010), method3 doesn't seem more accurate although it may be efficient. Besides, could the authors validate the extent to which the baselines derived using this algorithm represent the background?

Thirdly, many details about method3 were not shown in this paper but presented in Wang et al., 2018. I understand this is a method used in the published paper, but since this is a journal about measurement techniques, I think more details should be provided, especially the setting of the time window. Wang et al. (2018) used 8h, but is it appropriate here since a new station near a highway (NR-TOR-1) is added in this MS? As mentioned in L243-244, characteristics of emission sources determine the

frequency of signals. Thus should the time window for a station near highway may need to be different from that near streets, intersections or bus stops, which have their own frequency components (probably higher)? In addition, would different pollutant species require a different setting of window, especially a secondary pollutant such as ozone?

5) In L380-384, I don't understand why method1 and method3 are better as they both provide lower difference values. In my opinion, method3 has more disadvantages than method2, because the outcomes highly depend on the choice of time window and it's hard to determine if the baselines represent a real background.

As I understand it, method2 only used part of the data and clearly gave the largest difference between roadside and background concentrations. While the other two methods used the data even when the roadside stations experience background concentrations (e.g. under upwind conditions). Literally, the output from method2 is not the average local concentration. So I don't agree with the statement that method2 over-predicts average local concentrations (L131 in supplementary information). If the aim of this MS is determining the averaged concentration difference, method2 should be revised, otherwise, the difference between method2 and method1&3 is just caused by the difference in the methods.

6) L410-412, it seems the increase of pollutant concentrations under downwind conditions compared to upwind conditions is a main finding in this MS (as also mentioned in abstract). Could the authors provide the factors for each station and pollutant species? Theoretically, this factor should be a function of distance between source and receptor, wind speed, eddy diffusivity etc. Is it possible to add some tests about this?

7) In L484, why is method3 accurate and robust? Is it because the outputs agreed with those derived from method1?

References:

Liland, K. H., Almøy, T., & Mevik, B. H. (2010). Optimal choice of baseline correction for multivariate calibration of spectra. Applied spectroscopy, 64(9), 1007-1016.

Sabaliauskas, K., Jeong, C. H., Yao, X., & Evans, G. J. (2014). The application of wavelet decomposition to quantify the local and regional sources of ultrafine particles in cities. Atmospheric environment, 95, 249-257.

Xing, Y., Brimblecombe, P., & Ning, Z. (2019). Fine-scale spatial structure of air pollutant concentrations along bus routes. Science of the Total Environment, 658, 1-7.
* * *

---

## Referee Comment (RC3) · Anonymous Referee #3 · 2 May 2019

The authors report on near-road and background measurements of air pollution in Toronto and Vancouver, Canada. The manuscript includes an interesting and insightful comparison of methods for separating the local and background contributions to each pollutant measured at the near-road sites. Ozone and PM2.5 are identified as regional and mostly secondary and as a result their concentrations are not well-associated with local emissions on adjacent roadways. Carbon dioxide has high contributions to overall concentrations due to global background levels, though a contribution due to local traffic is still discernable at the near-road sites. The time series analysis/interpolation of windowed minima methods are a particularly interesting aspect of the manuscript. The manuscript is interesting and should be published in AMT after the authors consider and respond to the following comments.

Major Comments

1. In Tables 2-4, the authors should reorganize their presentation to show and directly compare results of all three methods for estimating local contributions ($C_L$) to measured concentrations at NR-TOR-1 (Table 2), NR-TOR-2 (Table 3), and NR-VAN (Table 4). The current organization of these tables emphasizes comparisons across the measurement sites, whereas the main point of the paper is to compare methods for estimating $C_L$.

2. I suggest the authors verify their regression coefficients relating pollutant concentrations to wind speed are consistent via separate analysis of weekday and weekend conditions: traffic conditions and emissions change on weekends, whereas average meteorology should be the same.

3. The presentation of NO/NO2 ratios is unconventional. I suggest reporting NO2/NOx instead, where NOx = NO + NO2. The reasons for variations in NO2/NOx among sites should consider differences in background ozone, transit/residence time in near-roadway setting, differences in diesel truck fractions (diesel has a higher NO2/NOx ratio in primary emissions). Also it appears the calibration of the chemiluminescent NOx analyzers was only checked regularly for NO. Was there any checking of NO2 converter efficiencies?

Minor Comments and Technical Corrections

Line 158, 193: minutely should be rewritten as one-minute

Line 242: many such algorithms (omit "of")

Line 302: non-tailpipe PM emissions such as brake and tire wear and road dust are expected to be predominantly in the coarse mode and should not contribute much to fine particle mass (PM2.5).

Lines 319-320: fix wording: the reason these values. . . is believed to be due the following reason

---

## Author Comment (AC1) · 25 Jun 2019

**Response to Anonymous Referee 1**

*Of course, it makes a big difference if a roadside station is located 6 m away from the nearest traffic line or 10 m. Actually, any different position may lead to different concentrations, because the variability of air pollutant concentrations varies largely spatially. The authors are aware of that. However, the tone sometimes suggests that the results (concentrations!) are transferable to other locations or situations. For example, the last sentence of the abstract ("Downwind conditions enhanced local concentrations by a factor of 2 relative to their mean, while upwind conditions suppressed them by a factor of 4") is, one the one hand, perfectly fine. On*

*the other hand, there is no caveat saying: "This applies to this very specific situation, don't generalize!" Also, referring to lines 289-299 and Table 2, the absolute number of $C_L$ are not comparable to each other between sites, even though drastic differences between the sites are apparent. The authors are asked to go through their manuscript and find more cautious wording in this respect.*

Reply: Thank you for underlining this point, and we agree that the wording should be as explicit as possible so as to not imply generalizability where it may not be applicable. The expectation with this analysis is that, since the local components of the concentrations are normalized with respect to their mean values, the shape of the curves of these normalized concentrations w.r.t. wind direction and wind speed ought to be somewhat generalizable for receptors near roadways with varying rates of emission. I.e., the areas above and below unity for these curves is always equivalent thanks to the following property:

$$\int_0^N \left( \frac{x(\theta)}{\bar{x}} - 1 \right) \cdot d\theta = \frac{1}{\bar{x}} \cdot \int_0^N x(\theta) \cdot d\theta - \int_0^N d\theta = \frac{N \cdot \bar{x}}{\bar{x}} - N = 0$$

However, as is pointed out, the distance of the receptor from the roadway along with the height of the sampling inlet will almost certainly impact the shapes of these curves, even if they are less impacted by source strength.

Action: The manuscript will be revised carefully to be as explicit as possible in its wording to properly address this caveat.

*In the eyes of this reviewer, the data set allows much more interesting analysis*

*of emission factors, for example between NOx and CO2. How do NOx/CO2 ratios or UFP/CO2 ratios compare o the results of similar studies? Similar applies to CO. It is however acknowledged that this is outside the scope of this study.*

Reply: Yes, we agree. In fact, emission factor analysis from this study has already been performed and reported by Wang et al. (2018) (see: doi.org/10.1021/acs.est.8b01914). This was the originally intended use for the background subtraction algorithm.

Action: The text will be updated to refer readers to this analysis.

*Section 4.3 and Table 4: Why not did authors apply his method for ozone? When using maxima instead of minima for the time series analysis this should be no problem to do. The justification given in lines 362-364 is no convincing. The urban background concentration of O3 could have been quantified that way, and be compared with the respective results of methods 1 and 2.*

Reply: This is a great suggestion and results will be updated accordingly to include it. For added simplicity, the same method of rolling minima interpolation can be applied to -1*O3(t), and once calculated the sign can be flipped again, thereby allowing the same algorithm to be applied. Figure 1 shows an example of this algorithm applied to O3 at NR-TOR-2. We can see that the difference between near-road O3 and inferred background is generally greatest when there are larger concentrations of NOx, which should be expected.

*Eq. 2 seems screwed. Probably, a parenthesis is missing on the right-hand side, opening before $C_{NR}[i]$ and closing after $C_{BG}[i]$.*

Reply: Agreed.

Action: Changed accordingly.

*Line 187: A justification should be given as of why M and N are typically not identical. It is a bit counterintuitive.*

Reply: The reason why M and N are typically not identical is that there will be a prevailing wind direction at most air quality monitoring locations. For example, if sampling is done continuously and data are not excluded based on wind direction, it should be expected that downwind data will occur more frequently than upwind data, for example, if it aligns with the prevailing wind direction of the site. The important point here is that N and M constitute sets of data that are inherently mutually exclusive (i.e. one cannot sample upwind and downwind of a road simultaneously with a single receptor) and may occur under different conditions (e.g. time of day).

Action: The text will be updated to elaborate on this point as it is perhaps not obvious in its current state. Further, wind rose plots will be included for each near-road location in the SI.

*Line 194: Why did you chose 75 % here? Likely, the results are more reliable if 100 % is used. See also line 413.*

Reply: Thank you for pointing this out. This was actually a mistake in the text. Hourly averages in general were included only if $\geq 75\%$ of minutely data were available, and this applied also to the meteorological data. For classifying whether a given hour was downwind or upwind, the hourly vector averages were used directly.

The only hours omitted were stagnant hours in which the wind speed was < 1.0 [m/s].

Action: The text will be corrected accordingly.

*The PM2.5 results are puzzling indeed. It could be the precision and accuracy of the analyzers not being able to resolve the small differences in concentrations between stations and within time series. If so, the results are not statistically robust. This issue should be analyzed in more detail and presented and discussed in the manuscript.*

Reply: You raise a good point regarding the precision between instruments, especially the PM2.5 monitors, and whether the background-subtraction methods are able to distinguish what is likely a very minor contribution to the total near-road signal. Regarding PM2.5 specifically, more in-depth results have been reported already by Sofowote et al. (2018) at NR-TOR-1 using an Aerodyne ACSM, XACT 625, AE33, and SHARP 5030, and their findings suggested the major component of PM2.5 responsible for these "local" fluctuations was black carbon, which is measured also using an AE33 here. Indeed, Table 2 would also suggest that BC is a major subset of this "local" PM2.5.

The SHARP 5030 manual specifies an hourly precision of "$\pm 2$ $\mu$g/m3 < 80 $\mu$g/m3; $\pm 5$ $\mu$g/m3 > 80 $\mu$g/m3", and a precision between two monitors of "$\pm 0.5$ $\mu$g/m3 (2-$\sigma$, 24-hour time resolution)" (Thermo Scientific, 2013). So, the average site differences between near-road and background sites, which are all around 2 $\mu$g/m3 or less and calculated over 2 years of data, are likely statistically significant results (Table 2). However, this raises an important issue as to why methods of background-subtraction applied to an hourly near-road time-series fails to properly pick out the local component: if the average local component is 2 $\mu$g/m3, and the hourly precision of the instrument is $\pm 2$ $\mu$g/m3, then the signal-to-noise ratio of this local component on an

hourly time scale is likely quite small and perhaps not detectable.

Action: This raises an important caveat regarding instrument precision limits and the application of background-subtraction algorithms to near-road data. Naturally, it brings to question the precision of all other instruments used in this study. Thus, instrumental precision will be reported in the methodology section, and reasons why the background subtraction algorithm for PM2.5 has seemingly failed will be discussed in the results.

Thank you for the feedback!

**References**

Sofowote, U. M., Healy, R. M., Su, Y., Debosz, J., Noble, M., Munoz, A., Jeong, C-H., Wang, J. M., Hilker, N., Evans, G. J., and Hopke, P. K.: Understanding the PM2.5 imbalance between a far and near-road location: Results of high temporal frequency source apportionment and parameterization of black carbon, Atmos. Env., 173, 277-288, doi:10.1016/j.atmosenv.2017.10.063, 2018.

Thermo Scientific Model 5030 SHARP Synchronized Hybrid Ambient, Real-time Particulate Monitor data sheet, Thermo Scientific, Obtained online: https://assets.thermofisher.com/TFS-Assets/LSG/Specification-Sheets/D19419 .pdf, 25JUN2019.

Wang, J. M., Jeong, C-H., Hilker, N., Shairsingh, K. K., Healy, R. M., Sofowote, U., Debosz, J., Su, Y., McGaughey, M., Doerksen, G., Munoz, T., White, L., Herod, D., and Evans, G. J.: Near-Road Air Pollutant Measurements: Accounting for Inter-Site Variability Using Emission Factors, Environ. Sci. Technol., 52, 9495-9504,

doi:10.1021/acs.est.8b01914, 2018.
* * *
[Figure]

**Fig. 1.** Example of background-subtraction algorithm applied to ambient O3 concentrations at the near-road downtown Toronto site, NR-TOR-2, wherein a rolling maximum is calculated rather than a rolling minimum

---

## Author Comment (AC2) · 1 Jul 2019

**Response to Anonymous Referee 2**

The authors are grateful for the useful comments and criticisms. Referee comments are shown in *italicized* text below, with author's replies and actions following each comment.

1) The first paragraph in Introduction about exposure is not that relevant to the rest of the paper, so would better focus on the topic of traffic-related pollutants close to the road.

Reply: Being that this is a methodology-focussed paper, and the primary interest is in better understanding traffic's contribution to traffic-related pollutant concentration in the near-road environment, we tend to agree with this comment. While the motivation for better understanding this is in part exposure-driven, it is not the main topic of the paper.

Action: The introduction will be altered to reduce emphasis on exposure/health effects and more on the methodology of characterizing near-road air pollution.

2) In L174, why the calculated average difference is expected to converge the true average difference between sites? Are the differences between sites normally distributed? Is this convergence non-trivial and not simply a property of averages or central limit theory?

Reply: Perhaps the wording in the manuscript is unnecessary and/or confusing in this section. The concept of random sampling and error theory has been addressed by others in the context of air quality monitoring (Xu et al., 2007), where the amount of data needed for a sample mean to converge to a "true" mean has been understood. The point to this statement was to imply a trivial convergence: as the number of samples increases so does the certainty in the mean of the difference (as a result of more variability due to seasonal effects, meteorology, etc.), similar to central limit theory as is pointed out.

Action: Wording of the manuscript in this section will be altered for greater clarity.

3) In L190, since the authors have realized the downwind and upwind scenarios may encompass different time frames and may influence the results, why not do some
Reply: Thank you for pointing out this lapse in analysis. Indeed, if downwind periods occur largely at night time, for example, then its average will be biased low. This is a relatively straight forward analysis and will be implemented in a revised version of the manuscript. As an example, refer to Fig. 1 in this document. At NR-TOR-1 downwind data are mostly uniformly distributed w.r.t. hour of day. Upwind conditions, however, are more likely to occur in the afternoon compared with the morning, meaning the upwind average will be defined by afternoon pollutant concentrations more so than morning concentration.

This is a potential issue as certain times of day will influence the mean values more so than others. One means of addressing this is to randomly sample an equivalent number of hours for each hour of day and compare the resulting distribution with the case in which all data is used to see if they are significantly different. As a proof of concept, observe the distributions of UFP concentrations at NR-TOR-1 in Figs. 2-3, which were generated by randomly sampling an equivalent number of points from each hour of the day (this number was determined from the minimum values in Fig. 1). The resulting downwind and upwind averages were  $5.64E+4 \pm 240$  [cm-3] and  $1.46E+4 \pm 260$  [cm-3] ( $\pm 1\sigma$ ), respectively (compare with values in Table 3 of the manuscript: 5.70E+4 and 1.53E+4, respectively). More rigorous statistical analyses will involve tests on whether these bootstrapped populations are significantly different than those reported in Table 3 of the manuscript.

Action: Additional analyses will be performed to create diurnal trends in frequency of downwind/upwind hours, as well as include rose wind plots for each near-road site (suggested also in response to RC1). Further, statistical tests will be performed between data sampled from a uniform distribution w.r.t. time of day vs. all data used.
**4) In L238-274, the method3 was not explained properly.**

Reply: While the algorithm has been described in detail already in Wang et al. (2018), we agree that a more mathematical description of the algorithm is necessary.

Action: The manuscript will be edited to include a more mathematically rigorous definition of the background-subtraction algorithm detailed in this section.

Firstly, 'Time-series analysis' seems too general and may not be a good subtitle here and in the rest part of the MS. It often implies decomposition and forecasting.

Reply: Agreed. Time-series analysis does seem too vague for this section, especially considering only one algorithm is really explored for signal deconvolution. Perhaps more appropriate is "baseline estimation" or "moving minimum" as you have suggested.

Action: The subtitle of this section will be changed to be more descriptive.

Secondly, the authors talked about the frequency of signals very often in the first two paragraphs (L238-254) and allude to the wavelet decomposition algorithm used by Sabaliauskas (2014) as similar to their method. But I think this is not quite right and misleading. What I expected after the description is a frequency analysis, but method 3 is approximately a 'moving minimum' baseline algorithm. As an example of signal processing and a spatial frequency domain in the road-environment can be seen in Xing and Brimblecombe (2019). Although wavelet analysis can also be used to exact baselines as shown by Liland et al. (2010), the underlying theory is different. There have been many baseline algorithms in Liland et al review (2010), method3 doesn't seem more accurate although it may be efficient. Besides, could the authors validate the extent to which the baselines derived using this algorithm represent the

AMTD
**background?**

Reply: Thank you for referencing these two articles. The Liland et al. (2010) review highlights many different algorithms intended for background subtraction of spectroscopic data that could be applied to pollutant time series for a similar effect. We agree that too much emphasis is placed on frequency analysis and, in particular, the discrete wavelet transform (which is not explored in this paper but is described as an analogue for what is instead used). While the notion of correlation between different spatial scales and signal frequencies is still relevant, we agree that the wording as it is does seem to imply that the results would contain some frequency-domain analysis, which was not the intention (these results would certainly be interesting but do not necessarily align with the scope of this manuscript).

In response to the last point, a validation of how well the derived baseline represents the urban background was performed already between NR-TOR-2 and BG-TOR-2 by Wang et al. (2018). We admit that this validation may need to be extended to NR-TOR-1 and NR-VAN, however. The intent behind comparing methods 1 and 3 was to validate that the differences as estimated using the background derived using method 3 was indeed similar to the differences measured between near-road and urban background stations pairs.

Action: Broader references to background-subtraction algorithms utilized by other atmospheric scientists will be included in the introduction, with emphasis being shifted away from frequency-domain analysis. Furthermore, the supplementary will be updated to include a table comparing the backgrounds determined by methods 1 and 3.

Thirdly, many details about method3 were not shown in the paper but presented in Wang et al., 2018. I understand this is a method in the published paper, but since this is a journal about measurement techniques, I think more details should be provided, especially the setting of the time window. Wang et al. (2018) used 8h, but
is it appropriate here since a new station near a highway (NR-TOR-1) is added in this MS? As mentioned in L243-244, characteristics of emission sources determine the frequency of signals. Thus should the time window for a station near highway may need to be different from that near streets, intersections or bus stops, which have their own frequency components (probably higher)? In addition, would different pollutant species require a different setting of window, especially a secondary pollutant such as ozone?

Reply: In response to the first point made regarding choice of time window and different receptors, while different near-road environments will inevitably affect higher frequency signals in a pollutant time-series (due to closer source proximities) as mentioned, the choice of time window was intended to be more of a reflection of the spatial scale differentiation between what is considered "background" and "local". As a first-order approximation, consider a primary pollutant affected only by physical dispersion. If this rate of dispersion is proportional to wind speed, then the pollutant's length of influence would in some way be proportional to:

 $d\approx u\cdot t$

where 'u' is wind speed and 't' is time since emission. Then, for a wind speed of 1.0 [m/s] and a time of 8 [hrs], for example, the pollutant's range of influence would be approximately 30 [km]. Thus, an argument could be made that utilizing a time window of 8 [hrs] in the background-subtraction algorithm is effectively distinguishing between emissions from sources within approximately 30 [km] of the receptor (those originating from nearest roadways will have the greatest influence on the signal) and those from outside of 30 [km]. Of course, this is a gross approximation and is likely not physically accurate, but it emphasizes the spatiotemporal relationship between signal frequency (choice of time window, which is related to signal cutoff frequency) and source distance. Moreover, given that these measurements were made within urban
regions with relative homogeneous distributions of roads, averaging the background over a smaller or larger spatial area, should not make much difference. Regarding the second point made (time window and different pollutants): you have raised an important issue, and we believe further analysis is necessary to support the time windows used in this study. A sensitivity analysis showing the distribution of measured urban background concentrations vs. the distribution of derived backgrounds as a function of time window, pollutant, and site would better support the time windows used in this study.

Action: A sensitivity analysis will be performed for each near-road/urban background station pair to compare measured urban background concentrations with derived baseline concentrations as a function of time window. This will further support the time windows used in this study and will determine whether different time windows for different pollutants is justified.

5) In L380-384, I don't understand why method1 and method3 are better as they both provide lower difference values. In my opinion, method3 has more disadvantages than method2, because the outcomes highly depend on the choice of time window and it's hard to determine if the baselines represent a real background.

Reply: The choice of wording here is perhaps inappropriate then, as the intention was not to claim one method being "better" than the other, but to highlight the advantages and disadvantages of each along with the fundamental differences between them. For example, methods 1 and 3 may be more appropriate for understanding traffic's influence to a 24 hour-averaged exposure from an epidemiological perspective (as all meteorological conditions are considered), whereas method2 may be better for extracting data whose impact from local traffic is greatest for use in fleet-averaged emission factor calculations, for example.
Action: This section will be reworded to include greater detail of the advantages and disadvantages of each method while retaining a tone of neutrality.

As I understand it, method2 only used part of the data and clearly gave the largest difference between roadside and background concentrations. While the other two methods used the data even when the roadside stations experience background concentrations (e.g. under upwind conditions). Literally, the output from method2 over-predicts average local concentrations (L131 in supplementary information). If the aim of this MS is determining the averaged concentration difference, method2 should be revised, otherwise, the difference between method2 and method13 is just caused by the difference in the methods.

Reply: We fully agree that what method 2 is measuring is inherently different from the other two methods. We will revise the text to better emphasise this important point. However, quantifying this difference is still of importance so that others might better understand the extent to which they differ. Moreover, method 2 provides an upper limit of the impact of the road on exposure. Section S3 proposes an alternative methodology for utilizing meteorological data that falls in line better with methods 1 and 3.

6) L410-412, it seems the increase of pollutant concentrations under downwind conditions compared to upwind conditions is a main finding in this MS (as also mentioned in abstract). Could the authors provide the factors for each station and pollutant species? Theoretically, this factor should be a function of distance between source and receptor, wind speed, eddy diffusivity etc. Is it possible to add some tests about this?
Reply: In the manuscript we have chosen to express this as the ratio of the local portion of the upwind and downwind concentration to the average value (Figure 4). This inherently makes more physical sense to use than directly comparing the ratio of the downwind to upwind concentrations, given that they both contain a "background" that is not related to the road. Values can be calculated and reported for each site and pollutant as a supplement to Figure 4, which just shows an agglomerated average for all species. Hypothetically, if these primary pollutants disperse similarly in the near-road regime and are not significantly impacted by secondary processes in the time it takes for them to be detected, then these curves should be similar between species. We agree that differences in dispersion between gas and particle-phase pollutants, and post-tailpipe transformation (e.g. UFP dynamics), for example, may lead to differences between pollutants.

The reason these trends were analysed with respect to normalized local concentrations was so that they would be invariant with respect to source strength. I.e., the area above and below unity for each curve are equivalent thanks to the property:

$$\int_0^N \left(\frac{x(\theta)}{\bar{x}} - 1\right) d\theta = 0$$

However, as you have mentioned, the shape will be impacted by distance of receptor to source, wind speed, eddy diffusivity, receptor height, atmospheric stability, etc. While we agree that the siting of these near-road stations along with meteorological conditions will have a theoretical impact on these data, it is out of the scope of this manuscript (the focus of which is a comparison of background subtraction methodologies) to attempt to model these results in a theoretical manner.

Action: Graphs similar to Figure 4 for each pollutant will be added to the SI, along with a table summarizing downwind/upwind ratios.
7) In L484, why is method3 accurate and robust? Is it because the outputs agreed with those derived from method1?

Reply: It is deemed accurate because it agrees with those values derived from method 1 which is the closest estimate to a real background. In terms of robustness, it appeared to be applicable across all near-road monitoring locations and data did not need to be filtered by meteorology, for example. How robust this algorithm is, exactly, will be available following the aforementioned time window sensitivity analysis.

Action: See above response regarding time window sensitivity analysis.

**References:**

Wang, J. M., Jeong, C-H., Hilker, N., Shairsingh, K. K., Healy, R. M., Sofowote, U., Debosz, J., Su, Y., McGaughey, M., Doerksen, G., Munoz, T., White, L., Herod, D., and Evans, G. J.: Near-Road Air Pollutant Measurements: Accounting for Inter-Site Variability Using Emission Factors, Environ. Sci. Technol., 52, 9495-9504, doi:10.1021/acs.est.8b01914, 2018.

Xu, X., Brook, J. R., and Guo, Y.: A Statistical Assessment of Saturation and Mobile Sampling Strategies to Estimate Long-Term Average Concentrations across Urban Areast, J. Air Waste. Manag. Assoc., 57, 1396-1406, doi:10.3155/1047-3289.57.11.1396, 2007.

**AMTD**
**Fig. 1.** Number of hours in which data were sampled downwind and upwind of Highway 401 at NR-TOR-1, aggregated by hour of day.
Interactive

comment

**Fig. 2.** Distribution of downwind UFP concentrations at NR-TOR-1, generated by bootstrapping (N = 100) an equivalent number of hours from e

---

## Author Comment (AC3) · 1 Jul 2019

**Response to Anonymous Referee 3**

The authors are grateful for the useful comments and technical corrections outlined here. Referee comments are shown in *italicized* text below, with author's replies and actions following each comment.

*Major comments*

*1. In Tables 2-4, the authors should reorganize their presentation to show and directly compare results of all three methods for estimating local contributions ($C_L$) to*

[Figure]

*measured concentration at NR-TOR-1 (Table 2), NR-TOR-2 (Table 3), and NR-VAN (Table 4). The current organization of these tables emphasizes comparisons across the measurement sites, whereas the main point of the paper is to compare methods for estimating $C_L$.*

Reply: We agree with this suggestion. Condensing the information into a singular table as you have suggested is likely the best way of presenting relevant information as efficiently as possible. Alternatively, the suggested table could be shown visually in a figure, in which the local concentrations determined by each method are compared between pollutants and sites.

Action: Tables 2-4 will be moved to the supplementary information, while the mean $C_L$ values from each will be agglomerated into a single table or figure to be included in the manuscript.

*2. I suggest the authors verify their regression coefficients relating pollutant concentrations to wind speed are consistent via separate analysis of weekday and weekend conditions: traffic conditions and emissions change on weekends, whereas average meteorology should be the same.*

Reply: This is a great suggestion and there is no reason not to include it in an updated manuscript version. As you have pointed out, since average meteorology should be similar between weekdays and weekends, regression between these two subsets should yield similar results. The primary difference between weekdays and weekends (aside from the frequency of data) are the volumes of traffic, which would yield greater local concentrations with respect to mean values, so the regression would effectively be modelling higher and lower ranges.

Action: The suggested analysis will be performed and included in the supplementary information.

*3. The presentation of NO/NO2 ratios is unconventional. I suggest reporting NO2/NOx instead, where NOx = NO + NO2. The reasons for variations in NO2/NOx among sites should consider differences in background ozone, transit/residence time in near-roadway setting, differences in diesel truck fractions (diesel has higher NO2/NOx ratio in primary emissions). Also it appears the calibration of the chemiluminescent NOx analyzers was only checked regularly for NO. Was there any checking of NO2 converter efficiencies?*

Reply: We agree that it is more sensible to instead report the ratio of NO2/NOx and will update the discussion of results in accordance with this.
Thank you for pointing out the converter efficiencies of the NOx analyzers. Indeed, the NO and NOx channels were calibrated using an NO standard located on-site. The manuscript needs to be updated to indicate that each station had either a Thermo 146i gas calibrator or an Environics 6100 multi-gas calibration system (only NR-TOR-2 used the Thermo). In addition to mixing various flow rates of zero and span gasses, these calibrators also have UV lamps, allowing O3 to be generated by a calibrated amount. This was how the O3 analyzers were calibrated. Additionally, following each NO/NOx calibration, a significant amount of O3 was generated (about 50% of NO by mole) to test the efficiency of the molybdenum converters. Generally, the efficiency of these converters was very close to 100%, and the test was only done to ensure a conversion efficiency of > 99.5%. The NO2 coefficients were left at 1.000, and if the instrument's converter looked like it was struggling (i.e. < 99.5%) then it was sent back to Thermo Scientific for calibration/maintenance. The fact that molybdenum converters were used is another important point as they cannot distinguish between NO2 and more oxidized forms of nitrogen: NOy (NOz – NOx). Being that local NO2 was defined

by short-term temporal fluctuations, however, it is doubtful that NOy (which is primarily affected by secondary chemistry) contributed to it substantially.

Action: Greater discussion of local quantities of NOx will be included in the results section, with ratios being reported as NO2/NOx rather than NO/NO2. Also, the methodology section will be updated to mention both calibrator models (Thermo 146i and Environics 6100) and converter efficiency checks for the 42i.

*Minor Comments and Technical Corrections*

*Line 158, 193: minutely should be rewritten as one-minute*

Reply: Changed.

*Line 242: many such algorithms (omit "of")*

Reply: Changed.

*Line 302: non-tailpipe PM emissions such as brake and tire wear and road dust are expected to be predominantly in the coarse mode and should not contribute much to fine particle mass (PM2.5).*

Reply: The text will be updated to emphasize this fact. While it is true that non-tailpipe emissions are generally greater than 2.5 microns in diameter, these sources still contribute enough to the PM2.5 size range to produce discernible differences between sites, and these differences are generally more heterogeneous than things such as secondary organics, for example (see Jeong et al., 2019).

[Figure]

*Lines 319-320: fix wording: the reason these values...is believed to be due the following reason*

Reply: This sentence will be reworded for clarity and brevity.

**References**:

Jeong, C-H., Wang, J. M., Hilker, N., Debosz, J., Sofowote, U., Su, Y., Noble, M., Healy, R. M., Munoz, T., Dabek-Zlotorzynska, E., Celo, V., White, L., Audette, C., Herod, D., and Evans, G. J.: Temporal and spatial variability of traffic-related PM2.5 sources: Comparison of exhaust and non-exhaust emissions, Atmos. Env., 198, 55-69, doi:10.1016/j.atmosenv.2018.10.038, 2019.

---

## Author Response (AR1)

**1 Author's Response**

Reviewer comments have been aggregated at the beginning of this marked-up manuscript, along with author's responses to them. Following these comments and responses is a summary of all changes to the most recent version of the manuscript.

Nathan Hilker nathan.hilker@mail.utoronto.ca

**7 Referee Comments and Responses**

Referee comments are displayed below in *italics* with author's responses following each comment. Actions taken are written
in **bold**.

**10 Anonymous Referee # 1**

This is a thorough application of three methods to distinguish traffic-related air pollutants from background concentrations, applied to three road-side stations in two cities in Canada. The results are novel, interesting, well presented. The manuscript 13 should be accepted for AMT after authors' response to the issues as raised below.

Of course, it makes a big difference if a roadside station is located 6 m away from the nearest traffic line or 10 m. Actually, any different position may lead to different concentrations, because the variability of air pollutant concentrations varies largely spatially. The authors are aware of that. However, the tone sometimes suggests that the results (concentrations!) are transferable to other locations or situations. For example, the last sentence of the abstract ("Downwind conditions enhanced local concentrations by a factor of 2 relative to their mean, while upwind conditions suppressed them by a factor of 4") is, one the one hand, perfectly fine. On the other hand, there is no caveat saying: "This applies to this very specific situation, don't generalize!" Also, referring to lines 289-299 and Table 2, the absolute number of CL are not comparable to each other between sites, even though drastic differences between the sites are apparent. The authors are asked to go through their manuscript and find more cautious wording in this respect.

Reply: Thank you for underlining this point, and we agree that the wording should be as explicit as possible so as to not imply generalizability where it may not be applicable. The expectation with this analysis is that, since the local components of the concentrations are normalized with respect to their mean values, the shape of the curves of these normalized concentrations w.r.t. wind direction and wind speed ought to be somewhat generalizable for receptors near roadways with varying rates of emission. I.e., the areas above and below unity for these curves is always equivalent thanks to the following property:

$$\int_{0}^{N} \left(\frac{x(\theta)}{x} - 1\right) \cdot d\theta = \frac{1}{x} \cdot \int_{0}^{N} x(\theta) \cdot d\theta - \int_{0}^{N} d\theta = \frac{N \cdot \bar{x}}{x} - N = 0$$
However, as is pointed out, the distance of the receptor from the roadway along with the height of the sampling inlet will almost certainly impact the shapes of these curves, even if they are less impacted by source strength.

Action:

L44-50: Abstract has been reworded to address this caveat.

Various parts of the manuscript also have been cautiously edited so as to not imply generalizability.

In the eyes of this reviewer, the data set allows much more interesting analysis of emission factors, for example between NOX and CO2. How do NOxCO2 ratios or UFP/CO2 ratios compare o the results of similar studies? Similar applies to CO. It is however acknowledged that this is outside the scope of this study.

Reply: Yes, we agree. In fact, emission factor analysis from this study has already been performed and reported by Wang et al. (2018) (see: doi.org/10.1021/acs.est.8b01914). This was the originally intended use for the background subtraction algorithm.

L206-298 edited to highlight this work for interest readers.

Section 4.3 and Table 4: Why not did authors apply his method for ozone? When using maxima instead of minima for the time series analysis this should be no problem to do. The justification given in lines 362-364 is no convincing. The urban background concentration of O3 could have been quantified that way, and be compared with the respective results of methods 1 and 2.

Reply: This is a great suggestion and results will be updated accordingly to include it. For added simplicity, the same method of of oiling minima interpolation can be applied to -1\*O3(t), and once calculated the sign can be flipped again, thereby allowing the same algorithm to be applied. Figure 1 shows an example of this algorithm applied to O3 at NR-TOR-2. We can see that the difference between near-road O3 and inferred background is generally greatest when there are larger concentrations of NOs, which should be expec

Added section 3.3.2 which explains how the methodology can be applied to ozone.

**61 Results in Tables 2-4 now include O3 concentrations.**

Eq. 2 seems screwed. Probably, a parenthesis is missing on the right-hand side, opening before CNR[i] and closing after
 CBG[i].

- 65
- 66 Reply: Agreed.
- 67
- 68 Action:
- 69 Eq. 2 updated.
- 70

Line 187: A justification should be given as of why M and N are typically not identical. It is a bit counterintuitive.

Reply: The reason why M and N are typically not identical is that there will be a prevailing wind direction at most air quality 74 monitoring locations. For example, if sampling is done continuously and data are not excluded based on wind direction, it 75 should be expected that downwind data will occur more frequently than upwind data, for example, if it aligns with the 76 prevailing wind direction of the site. The important point here is that N and M constitute sets of data that are inherently mutually exclusive (i.e. one cannot sample upwind and downwind of a road simultaneously with a single receptor) and may occur under different conditions (e.g. time of day).

Action:

**81 L222-229 updated for greater clarity as to why M and N are not typically identical.**

Line 194: Why did you chose 75 % here? Likely, the results are more reliable if 100 % is used. See also line 413.

Reply: Thank you for pointing this out. This was actually a mistake in the text. Hourly averages in general were included only if  $\geq$  75% of minutely data were available, and this applied also to the meteorological data. For classifying whether a given hour was downwind or upwind, the hourly vector averages were used directly. The only hours omitted were stagnant hours in which the wind speed was < 1.0 [m/s].

**90 Action: Section 3.2 has been updated to remove sentences referring to this.**

**92 The PM2.5 results are puzzling indeed. It could be the precision and accuracy of the analyzers not being able to resolve the**

small differences in concentrations between stations and within time series. If so, the results are not statistically robust. This issue should be analyzed in more detail and presented and discussed in the manuscript.

Reply: You raise a good point regarding the precision between instruments, especially the PM2.5 monitors, and whether the 97 background-subtraction methods are able to distinguish what is likely a very minor contribution to the total near-road signal. 98 Regarding PM2.5 specifically, more in-depth results have been reported already by Sofowote et al. (2018) at NR-TOR-1 using 99 an Aerodyne ACSM, XACT 625, AE33, and SHARP 5030, and their findings suggested the major component of PM2.5 100 responsible for these "local" fluctuations was black carbon, which is measured also using an AE33 here. Indeed, Table 2 would 101 also suggest that BC is a major subset of this "local" PM2.5.

The SHARP 5030 manual specifies an hourly precision of " $\pm 2 \ \mu g/m3 < 80 \ \mu g/m3$ ;  $\pm 5 \ \mu g/m3 > 80 \ \mu g/m3$ ", and a precision between two monitors of " $\pm 0.5 \ \mu g/m3$  (2- $\sigma$ , 24-hour time resolution)" (Thermo Scientific, 2013). So, the average site differences between near-road and background sites, which are all around 2  $\mu g/m3$  or less and calculated over 2 years of data, are likely statistically significant results (Table 2). However, this raises an important issue as to why methods of backgroundsubtraction applied to an hourly near-road time-series fails to properly pick out the local component: if the average local component is 2  $\mu g/m3$ , and the hourly precision of the instrument is  $\pm 2 \ \mu g/m3$ , then the signal-to-noise ratio of this local component on an hourly time scale is likely quite small and perhaps not detectable.

**110 Action:**

L493-495 added to address this issue.

**113 References**

Sofowote, U. M., Healy, R. M., Su, Y., Debosz, J., Noble, M., Munoz, A., Jeong, C-H., Wang, J. M., Hilker, N., Evans, G. J.,
and Hopke, P. K.: Understanding the PM2.5 imbalance between a far and near-road location: Results of high temporal
frequency source apportionment and parameterization of black carbon, Atmos. Env., 173, 277-288,
doi:10.1016/j.atmosenv.2017.10.063, 2018.

Thermo Scientific Model 5030 SHARP Synchronized Hybrid Ambient, Realtime Particulate Monitor data sheet, Thermo
 Scientific, Obtained online: https://assets.thermofisher.com/TFS-Assets/LSG/Specification-Sheets/D19419.pdf, 25JUN2019.

Wang, J. M., Jeong, C-H., Hilker, N., Shairsingh, K. K., Healy, R. M., Sofowote, U., Debosz, J., Su, Y., McGaughey, M.,

Doerksen, G., Munoz, T., White, L., Herod, D., and Evans, G. J.: Near-Road Air Pollutant Measurements: Accounting for

Inter- Site Variability Using Emission Factors, Environ. Sci. Technol., 52, 9495-9504, doi:10.1021/acs.est.8b01914, 2018.

Fig. 1. Example of background-subtraction algorithm applied to ambient O3 concentrations at the near-road downtown Toronto 128 site, NR-TOR-2, wherein a rolling maximum is calculated rather than a rolling minimum

**139 Anonymous Referee # 2**

| 140 | 1) The first paragraph in Introduction about exposure is not that relevant to the rest of the paper, so would better focus on the  |
|-----|------------------------------------------------------------------------------------------------------------------------------------|
| 141 | topic of traffic-related pollutants close to the road.                                                                             |
| 142 |                                                                                                                                    |
| 143 | Reply: Being that this is a methodology-focussed paper, and the primary interest is in better understanding traffic's contribution |
| 144 | to traffic-related pollutant concentration in the near-road environment, we tend to agree with this comment. While the             |
| 145 | motivation for better understanding this is in part exposure-driven, it is not the main topic of the paper.                        |
| 146 |                                                                                                                                    |
| 147 | Action:                                                                                                                            |
| 148 | Much of the first paragraph has been removed (i.e. those parts pertaining to health effects).                                      |
| 149 |                                                                                                                                    |
| 150 | 2) In L174, why the calculated average difference is expected to converge the true average difference between sites? Are the       |
| 151 | differences between sites normally distributed? Is this convergence non-trivial and not simply a property of averages or central   |
| 152 | limit theory?                                                                                                                      |
| 153 |                                                                                                                                    |
| 154 | Reply: Perhaps the wording in the manuscript is unnecessary and/or confusing in this section. The concept of random sampling       |
| 155 | and error theory has been addressed by others in the context of air quality monitoring (Xu et al., 2007), where the amount of      |
| 156 | data needed for a sample mean to converge to a "true" mean has been understood. The point to this statement was to imply a         |
| 157 | trivial convergence: as the number of samples increases so does the certainty in the mean of the difference (as a result of more   |
| 158 | variability due to seasonal effects, meteorology, etc.), similar to central limit theory as is pointed out.                        |
| 159 |                                                                                                                                    |
| 160 | Action:                                                                                                                            |
| 161 | L199-202: wording has been altered to better clarify this point.                                                                   |
| 162 |                                                                                                                                    |
| 163 | 3) In L190, since the authors have realized the downwind and upwind scenarios may encompass different time frames and may          |
| 164 | influence the results, why not do some statistical tests? It seems important to the final outcomes.                                |
| 165 |                                                                                                                                    |
| 166 | Reply: Thank you for pointing out this lapse in analysis. Indeed, if downwind periods occur largely at night time, for example,    |
| 167 | then its average will be biased low. This is a relatively straight forward analysis and will be implemented in a revised version   |
| 168 | of the manuscript. As an example, refer to Fig. 1 in this document. At NR-TOR-1 downwind data are mostly uniformly                 |
| 169 | distributed w.r.t. hour of day. Upwind conditions, however, are more likely to occur in the afternoon compared with the            |
| 170 | morning, meaning the upwind average will be defined by afternoon pollutant concentrations more so than morning                     |

concentration.

| 172 | This is a potential issue as certain times of day will influence the mean values more so than others. One means of addressing                 |
|-----|-----------------------------------------------------------------------------------------------------------------------------------------------|
| 173 | this is to randomly sample an equivalent number of hours for each hour of day and compare the resulting distribution with the                 |
| 174 | case in which all data is used to see if they are significantly different. As a proof of concept, observe the distributions of UFP            |
| 175 | concentrations at NR-TOR-1 in Figs. 2-3, which were generated by randomly sampling an equivalent number of points from                        |
| 176 | each hour of the day (this number was determined from the minimum values in Fig. 1). The resulting downwind and upwind                        |
| 177 | averages were 5.64E+4 $\pm$ 240 [cm-3] and 1.46E+4 $\pm$ 260 [cm-3] ( $\pm$ 1 $\sigma$ ), respectively (compare with values in Table 3 of the |
| 178 | manuscript: 5.70E+4 and 1.53E+4, respectively). More rigorous statistical analyses will involve tests on whether these                        |
| 179 | bootstrapped populations are significantly different than those reported in Table 3 of the manuscript.                                        |
| 180 |                                                                                                                                               |
| 181 | Action:                                                                                                                                       |
| 182 | L427-429 address this issue.                                                                                                                  |
| 183 | Fig. S2 added to the supplementary information.                                                                                               |
| 184 | Tables S5 and S6 added to the supplementary information.                                                                                      |
| 185 |                                                                                                                                               |
| 186 | 4) In L238-274, the method3 was not explained properly.                                                                                       |
| 187 |                                                                                                                                               |
| 188 | Reply: While the algorithm has been described in detail already in Wang et al. (2018), we agree that a more mathematical                      |
| 189 | description of the algorithm is necessary.                                                                                                    |
| 190 |                                                                                                                                               |
| 191 | Action:                                                                                                                                       |
| 192 | Section 3.3.1 has been almost entirely updated so as to describe the algorithm with greater mathematical rigor.                               |
| 193 |                                                                                                                                               |
| 194 | Firstly, 'Time-series analysis' seems too general and may not be a good subtitle here and in the rest part of the MS. It often                |
| 195 | implies decomposition and forecasting.                                                                                                        |
| 196 |                                                                                                                                               |
| 197 | Reply: Agreed. Time-series analysis does seem too vague for this section, especially considering only one algorithm is really                 |
| 198 | explored for signal deconvolution. Perhaps more appropriate is "baseline estimation" or "moving minimum" as you have                          |
| 199 | suggested.                                                                                                                                    |
| 200 |                                                                                                                                               |
| 201 | Action:                                                                                                                                       |
| 202 | Section 3.3 title changed to "Background subtraction using time series data"                                                                  |
| 203 |                                                                                                                                               |
| 204 | Secondly, the authors talked about the frequency of signals very often in the first two paragraphs (L238-254) and allude to the               |

wavelet decomposition algorithm used by Sabaliauskas (2014) as similar to their method. But I think this is not quite right and misleading. What I expected after the description is a frequency analysis, but method 3 is approximately a 'moving minimum'

baseline algorithm. As an example of signal processing and a spatial frequency domain in the road-environment can be seen in Xing and Brimblecombe (2019). Although wavelet analysis can also be used to exact baselines as shown by Liland et al.

(2010), the underlying theory is different. There have been many baseline algorithms in Liland et al review (2010), method3

doesn't seem more accurate although it may be efficient. Besides, could the authors validate the extent to which the baselines derived using this algorithm represent the background?

**213 Action:**

Wording in the introduction and Section 3.3.1 has been changed to avoid any emphasis on frequency-domain analysis.
To address the second point, Section S4 has been added to the supplementary and is discussed in Section 4.3 of the manuscript.

218Thirdly, many details about method3 were not shown in the paper but presented in Wang et al., 2018. I understand this is a219method in the published paper, but since this is a journal about measurement techniques, I think more details should be220provided, especially the setting of the time window. Wang et al. (2018) used 8h, but is it appropriate here since a new station221near a highway (NR-TOR-1) is added in this MS? As mentioned in L243-244, characteristics of emission sources determine222the frequency of signals. Thus should the time window for a station near highway may need to be different from that near223streets, intersections or bus stops, which have their own frequency components (probably higher)? In addition, would different224pollutant species require a different setting of window, especially a secondary pollutant such as ozone?

Reply: In response to the first point made regarding choice of time window and different receptors, while different near-road environments will inevitably affect higher frequency signals in a pollutant time-series (due to closer source proximities) as mentioned, the choice of time window was intended to be more of a reflection of the spatial scale differentiation between what is considered "background" and "local". As a first-order approximation, consider a primary pollutant affected only by physical dispersion. If this rate of dispersion is proportional to wind speed, then the pollutant's length of influence would in some way be proportional to:

**$d \approx u \cdot t$**

where 'u' is wind speed and 't' is time since emission. Then, for a wind speed of 1.0 [m/s] and a time of 8 [hrs], for example, the pollutant's range of influence would be approximately 30 [km]. Thus, an argument could be made that utilizing a time window of 8 [hrs] in the background-subtraction algorithm is effectively distinguishing between emissions from sources within approximately 30 [km] of the receptor (those originating from nearest roadways will have the greatest influence on the signal) and those from outside of 30 [km]. Of course, this is a gross approximation and is likely not physically accurate, but it emphasizes the spatiotemporal relationship between signal frequency (choice of time window, which is related to signal cutoff frequency) and source distance. Moreover, given that these measurements were made within urban regions with relative 240 homogeneous distributions of roads, averaging the background over a smaller or larger spatial area, should not make much 241 difference.

Regarding the second point made (time window and different pollutants): you have raised an important issue, and we believe further analysis is necessary to support the time windows used in this study. A sensitivity analysis showing the distribution of

 $244 \qquad \text{measured urban background concentrations vs. the distribution of derived backgrounds as a function of time window, pollutant,}$

and site would better support the time windows used in this study.

**247 Action:**

L496-526 in Section 4.3 have been updated to discuss the point regarding the choice of time window for method 3.

5) In L380-384, I don't understand why method1 and method3 are better as they both provide lower difference values. In my 251 opinion, method3 has more disadvantages than method2, because the outcomes highly depend on the choice of time window 252 and it's hard to determine if the baselines represent a real background.

Reply: The choice of wording here is perhaps inappropriate then, as the intention was not to claim one method being "better" than the other, but to highlight the advantages and disadvantages of each along with the fundamental differences between them. For example, methods 1 and 3 may be more appropriate for understanding traffic's influence to a 24 hour-averaged exposure from an epidemiological perspective (as all meteorological conditions are considered), whereas method2 may be better for extracting data whose impact from local traffic is greatest for use in fleet-averaged emission factor calculations, for example.

Action:

**261 L534-538: wording updated to better discuss the pros and cons of the methods used.**

**As I understand it, method2 only used part of the data and clearly gave the largest difference between roadside and background concentrations. While the other two methods used the data even when the roadside stations experience background concentrations (e.g. under upwind conditions). Literally, the output from method2 over-predicts average local concentrations (L131 in supplementary information). If the aim of this MS is determining the averaged concentration difference, method2 should be revised, otherwise, the difference between method2 and method13 is just caused by the difference in the methods.**

Reply: We fully agree that what method 2 is measuring is inherently different from the other two methods. We will revise the text to better emphasise this important point. However, quantifying this difference is still of importance so that others might better understand the extent to which they differ. Moreover, method 2 provides an upper limit of the impact of the road on exposure. Section S3 proposes an alternative methodology for utilizing meteorological data that falls in line better with methods 1 and 3.

**275 Action: see previous.**

6) L410-412, it seems the increase of pollutant concentrations under downwind conditions compared to upwind conditions is a main finding in this MS (as also mentioned in abstract). Could the authors provide the factors for each station and pollutant species? Theoretically, this factor should be a function of distance between source and receptor, wind speed, eddy diffusivity etc. Is it possible to add some tests about this?

Reply: In the manuscript we have chosen to express this as the ratio of the local portion of the upwind and downwind 282 concentration to the average value (Figure 4). This inherently makes more physical sense to use than directly comparing the 283 ratio of the downwind to upwind concentrations, given that they both contain a "background" that is not related to the road. 284 Values can be calculated and reported for each site and pollutant as a supplement to Figure 4, which just shows an agglomerated 285 average for all species. Hypothetically, if these primary pollutants disperse similarly in the near-road regime and are not 286 significantly impacted by secondary processes in the time it takes for them to be detected, then these curves should be similar 287 between species. We agree that differences in dispersion between gas and particle-phase pollutants, and post-tailpipe transformation (e.g. UFP dynamics), for example, may lead to differences between pollutants. 288

The reason these trends were analysed with respect to normalized local concentrations was so that they would be invariant with respect to source strength. I.e., the area above and below unity for each curve are equivalent thanks to the property:

$$\int_{0}^{N} \left( \frac{x(\theta)}{\bar{x}} - 1 \right) d\theta = 0$$

However, as you have mentioned, the shape will be impacted by distance of receptor to source, wind speed, eddy diffusivity, receptor height, atmospheric stability, etc. While we agree that the siting of these near-road stations along with meteorological conditions will have a theoretical impact on these data, it is out of the scope of this manuscript (the focus of which is a comparison of background subtraction methodologies) to attempt to model these results in a theoretical manner.

7) In L484, why is method3 accurate and robust? Is it because the outputs agreed with those derived from method1?298

Reply: It is deemed accurate because it agrees with those values derived from method 1 which is the closest estimate to a real background. In terms of robustness, it appeared to be applicable across all near-road monitoring locations and data did not need to be filtered by meteorology, for example. How robust this algorithm is, exactly, will be available following the aforementioned time window sensitivity analysis.

Action: See above response regarding time window sensitivity analysis.

**305 References**

Wang, J. M., Jeong, C-H., Hilker, N., Shairsingh, K. K., Healy, R. M., Sofowote, U., Debosz, J., Su, Y., McGaughey, M.,
Doerksen, G., Munoz, T., White, L., Herod, D., and Evans, G. J.: Near-Road Air Pollutant Measurements: Accounting for
Inter- Site Variability Using Emission Factors, Environ. Sci. Technol., 52, 9495-9504, doi:10.1021/acs.est.8b01914, 2018.

Xu, X., Brook, J. R., and Guo, Y.: A Statistical Assessment of Saturation and Mobile Sampling Strategies to Estimate Long-

Term Average Concentrations across Urban Areast, J. Air Waste. Manag. Assoc., 57, 1396-1406, doi:10.3155/1047 3289.57.11.1396, 2007.

day.

Fig. 2. Distribution of downwind UFP concentrations at NR-TOR-1, generated by bootstrapping (N = 100) an equivalent number of

Fig. 2. Distribution of downwirhours from each hour of day.